# Study on the Coupled Vibration Characteristics of a Two-Stage Bladed Disk Rotor System

**Yinxin Yu** [1], **Xiaolong Jin** [1], **Yanming Fu** [2,*] and **Tianyu Zhao** [3,*]

1   Faculty of Mechanical and Electrical Engineering, Kunming University of Science and Technology, Kunming 650500, China; yxy@kust.edu.cn (Y.Y.); jinxiaolong@stu.kust.edu.cn (X.J.)
2   Laboratory Management Center, Shenyang Sport University, Shenyang 110102, China
3   School of Science, Northeastern University, Shenyang 110819, China
*   Correspondence: fuym@syty.edu.cn (Y.F.); zhaotianyu@mail.neu.edu.cn (T.Z.)

**Abstract:** This paper conducts a coupled vibration analysis of a two-stage bladed disk rotor system. According to the finite element method, the bladed disk rotor system is established. The substructure modal synthesis super-element method (SMSM) with a fixed interface and free interface is presented to obtain the vibration behaviors of the rotor system. Then, the free vibration results are compared with the ones calculated by the cyclic symmetry analysis method to validate the analysis in this paper. The results show that the modes of the two-stage bladed disk not only include the modes of the first- and second-stage bladed disk, but also the coupled modes of the two-stage bladed disk.

**Keywords:** bladed disk; modal synthesis super-element method; coupled vibration; rotor

## 1. Introduction

An aero-engine is a kind of high-speed rotating machinery with a complex structure. The rotating blades and the fixed bladed disk are the important key parts of the aero-engine. In an aero-engine, multi-stage bladed disks are assembled together, and the study of the coupling interaction between the multi-stage bladed disks is particularly important for understanding the dynamics characteristics of the whole engine. In the vibration analysis of the bladed disk, the vibration coupling form between the blade and disk is usually analyzed. However, the interstage coupled effect is usually ignored. For the multi-stage bladed disk system, the coupling between stages is an important factor affecting the energy propagation between the disks. The multi-stage rotor has a specific type of mode and response mode that extends a multi-stage bladed disk structure. Therefore, it is very important to analyze the interstage coupled vibration of the multi-stage bladed disk, which is also the basis for further study of the interstage coupled vibration caused by the mistuned single-stage bladed disk.

In recent years, scholars have carried out theoretical, numerical simulation, and experimental studies on the dynamic characteristics of the bladed disk. In view of the actual structure in engineering, the finite element method is usually used to model and analyze the complex bladed disk. For the multi-stage and multi-component integral bladed disk assemblies, potential topics have been proposed in [1]; for instance, building a more effective and applicative model with higher precision. Based on the Timoshenko beam theory and Kirchhoff plate theory, Laxalde et al. [2] propose a new method that combines the cyclic modelling of each stage with a realistic inter-stage coupling. Study cases are presented to evaluate the efficiency of the method. Joannin et al. [3] introduces a novel reduced-order modelling technique well-suited to the study of nonlinear vibrations in large finite element models. The performance of the method is appraised on a nonlinear finite element model of the bladed disk in the presence of structural mistuning. HS et al. [4] proposed an improved shaft–disk–blade coupling model to study the influence of disk position and flexibility on critical speed and natural frequency in the coupling-disk-blade

unit. Zhao et al. [5] established the finite element model of the fully flexible shaft-disk-sleeve system with tip rub fault by using the Lagrange multiplier method, and proposed an improved disk-blade interface coupling method. Ma et al. [6] established a rotor-blade system dynamics model. With the increase in the number of blades, complex coupling modes, such as the vane-blade coupling mode, rotor lateral vibration and blade bending coupling mode, and rotor torsional vibration and blade bending coupling mode, have also appeared. Zhao et al. [7] established the coupled model of spinning shaft-disk assemblies under sliding bearing supports. For the multi-stage assembly of cyclic structures, Wang et al. [8] simulated and analyzed the vibration characteristics of ceramic matrix composite monolayers and found that the blade had a great influence on the vibration mode of the entire blade disc. On the basis of previous studies, Tang et al. [9] further established a vane-disk-shaft coupling model and explained the influence of tuning/detuning on the coupling modal characteristics. Al Bedoor B.O. [10] established a mathematical model of reduced order and studied the natural frequencies of shaft-torsion and blade-bending coupling. Huang et al. [11] studied axial-torsional, disk-lateral, and blade-bending coupled vibrations in a shaft–disk–blade unit. Chiu et al. [12] investigated the influence on the coupling vibrations among shaft-torsion and blade-bending coupling vibrations of a multi-disk rotor system. Chiu et al. [13] studied the influence of shaft torsion, blade bending, and wire-drawing coupling vibration on the coupled vibration of a multi-disk rotor system with group blades. Ma et al. [14] analyzed the effects of blade stagger angles on the blade rubbing-induced responses of a rotational shaft–disk–blade system. Wang et al. [15] analyzed the nonlinear dynamic behavior of a rotor-bearing system with interaction between the blades and rotor. Luo et al. [16] investigated the natural frequency of the free transverse vibration of blades in rotating disks to examine the relationship of the natural frequencies, blade stiffness, and nodal diameters, to study how neighboring blades react upon each other and affect the blade's natural frequency. Rzadkowski et al. [17] adopted the forced vibration analysis method and considered the influence of multistage coupling on the dynamic characteristics of an octave-disc rotor on a solid shaft. The results show that multi-level coupling must be considered in the design of rotor blades and discs to avoid the resonance caused by low-frequency flow excitation. Bladh et al. [18] studied the influence of interstage coupling on the dynamic performance of harmonic and detuned multistage blade-disk structures and pointed out that the dynamic performance of a single-stage rotor depends on the selection of the interstage coupled boundary conditions. Based on sector mistuning, Vargiu et al. [19] established a reduced order model for the dynamic analysis of mistuned bladed disks. Sector frequency mistuning is preferable to capture blade-to-disk irregularities. Petrov et al. [20] proposed an efficient method for analysis of nonlinear vibrations of mistuned bladed disk assemblies. For a practical high-pressure bladed turbine disk, considering several types of nonlinear forced response, the analysis of nonlinear forced response for simplified and realistic models of mistuned bladed disks has been performed. Rzadkowski et al. [21] studied the forced vibration of eight detuned blade discs on a solid shaft and found that when the blade disc was on the shaft, detuning had little influence on the blade stress. Chaofeng Li et al. [22] studied the coupling vibration characteristics of a flexible shaft–disk–blade system with detuning characteristics. Due to the detuning characteristics, the natural frequency and coupling mode types will change accordingly. Huang et al. [23] used a disk comprising of periodically shrouded blades to simulate the weakly coupled periodic structure. The effects of Coriolis force and the magnitude of disorder on the localization phenomenon of a rotating blade-disk system were investigated numerically. Zhao et al. [24–34] studied the vibration characteristics of a graphene nanoplatelet (GPL)-reinforced blade-disk rotor system by the experimental method and the finite element (FE) method, and studied the parallel intelligent algorithm based on a computed unified device architecture. The genetic particle swarm optimization algorithm is used for optimization arrangement on mistuned blades. The above works are mainly based on the finite element method to study the coupled modes and responses of the shaft–disk–blades system under the tuned and mistuned bladed disk. The coupled

modes of the multi-stage blade-disc system have not been studied, and the coupled model is greatly simplified compared with the actual structure.

In this paper, respectively using the substructure modal synthesis super-element method and the cyclic symmetry analysis method, two kinds of accurate finite element model for two-stage bladed disk were established. The accuracy of the substructure modal synthesis super-element method was verified by the cyclic symmetry analysis method. The interstage coupled vibration of the two-stage bladed disk was analyzed. This research fills in the blanks regarding the interstage coupled vibration of a complex bladed disk and lays a foundation for further research on the effect of a mistuned bladed disk on interstage coupled vibration.

## 2. Materials and Methods

Using the fixed interface prestress-free interface substructure modal synthesis super-element method, based on the finite element analysis software ANSYS, the dynamic frequency analysis of the first-stage bladed disk system of the compressor was carried out. The finite element model of the basic sector is shown in Figure 1. The parameters of the blade tenon and tenon grooves are respectively as follows: elastic modulus $E_0 = 1.135 \times 10^{11}$ Pa, Poisson's ratio $\mu_0 = 0.3$, and density $\rho_0 = 4380 \text{kg/m}^3$. The material parameters of the disk are as follows: elastic modulus $E_1 = 1.15 \times 10^{11}$ Pa, Poisson's ratio $\mu_1 = 0.3$, density $\rho_1 = 4640 \text{kg/m}^3$, and the contact form of the blade tenon and tenon grooves adopts a standard contact.

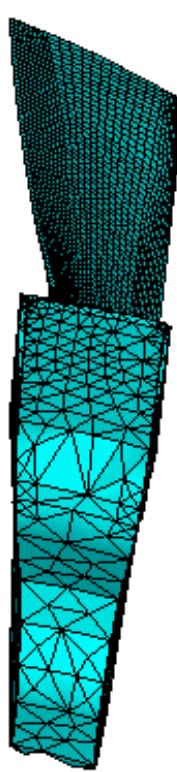

**Figure 1.** Finite element model of the basic sector.

The analysis process of the substructure modal synthesis super-element method is shown in Figure 2. For the modal synthesis super-element method, for the fixed interface prestressed and free interface substructures, the basic idea is that the finite element model of the basic sector of the bladed disk is established by using the substructure analysis method from bottom to top. The two side outlet degrees of freedom (master degrees of freedom) of the basic sectors of the bladed disk are fixed and the working speed is applied to perform the prestressed contact analysis (bladed binding, bladed contact) for each basic

sector of the detuned bladed disk. One opens the prestress setting and releases the fixed constraints of the two side exit degrees of freedom (master degrees of freedom) of the basic sector of the bladed disk, and then conducts the modal synthesis generation part analysis of the substructure of the free interface. One then generates a supercell and use supercell nesting technology to generate a multilevel supercell to complete the generation part. Secondly, the superunits are connected to analyze the overall bladed disk system (modal, dynamic response), and the use part is completed. Finally, the condensed solution of the dynamic response of the supercell master degree of freedom is extended to all the degrees of freedom in the supercell, so as to obtain the complete solution of the dynamic response of all the degrees of freedom in the bladed disk system, completing the extension part.

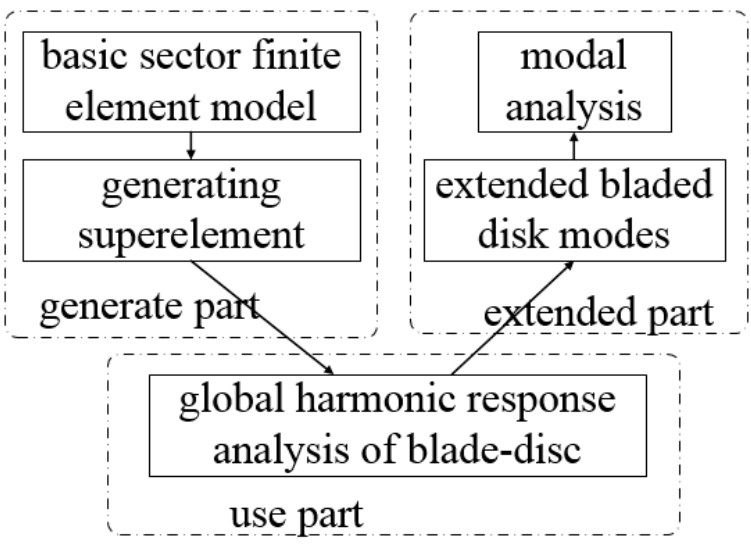

**Figure 2.** Analysis process of SMSM.

*2.1. Super-Element Power Reduction*

The motion equation of the superunit with interfacial force is

$$\begin{bmatrix} m_{ii} & m_{ij} \\ m_{ji} & m_{jj} \end{bmatrix} \left\{ \begin{array}{c} \ddot{x}_i \\ \ddot{x}_j \end{array} \right\} + \begin{bmatrix} k_{ii} & k_{ij} \\ k_{ji} & k_{jj} \end{bmatrix} \left\{ \begin{array}{c} x_i \\ x_j \end{array} \right\} = \left\{ \begin{array}{c} f_i \\ 0 \end{array} \right\} \tag{1}$$

where $x_i$ is the displacement of interface nodes, which is also the coordinate of main degrees of freedom; $x_j$ is the displacement of internal nodes, namely, the coordinate of deputy degrees of freedom; and $f_i$ is the interface strength.

Assuming that $\{x\} = e^{jwt}\{x\}$ and $\{f_i\} = e^{jwt}\{F_i\}$ and $\omega^2 = \lambda$, then, it can be obtained by Equation (1).

$$\begin{bmatrix} k_{ii} & k_{ij} \\ k_{ji} & k_{jj} \end{bmatrix} \left\{ \begin{array}{c} x_i \\ x_j \end{array} \right\} - \lambda \begin{bmatrix} m_{ii} & m_{ij} \\ m_{ji} & m_{jj} \end{bmatrix} \left\{ \begin{array}{c} x_i \\ x_j \end{array} \right\} = \left\{ \begin{array}{c} F_i \\ 0 \end{array} \right\} \tag{2}$$

assuming that $\lambda = \omega^2$, it can be obtained from Equation (2).

$$\{x_j\} = [\beta][x_i] \tag{3}$$

where $[\beta] = -\left[k_{jj} - \omega^2 m_{ij}\right]^{-1}\left[k_{ij} - \omega^2 m_{ji}\right]$, then

$$\{x\} = \left\{ \begin{array}{c} x_i \\ x_j \end{array} \right\} = \begin{bmatrix} I \\ \beta \end{bmatrix} \{x_i\} = [\beta_1][x_i] \tag{4}$$

Substitute this equation into Equation (2).

$$\left[\overline{k}\right]\{x_i\} - \lambda[\overline{m}]\{x_i\} = \{\overline{F}_i\} \tag{5}$$

where

$$\begin{cases} \left[\overline{k}\right] = [\beta_1]^T \begin{bmatrix} k_{ii} & k_{ij} \\ k_{ji} & k_{jj} \end{bmatrix} [\beta_1] \\ [\overline{m}] = [\beta_1]^T \begin{bmatrix} m_{ii} & m_{ij} \\ m_{ji} & m_{jj} \end{bmatrix} [\beta_1] \\ \overline{F} = [\beta_1]^T [F_i] \end{cases}$$

To constrain the degree of freedom of the interface, namely $\{x_j\} = 0$, can be obtained from the second equation in Equation (2).

$$\left[k_{jj}\right]\{x_j\} - \lambda\left[m_{jj}\right]\{x_j\} = 0 \tag{6}$$

by this formula, the main mode $[\Phi]$ of the fixed interface is obtained and regularized, and then

$$\begin{cases} [\Phi]^T [m_{jj}][\Phi] = [I] \\ [\Phi]^T [k_{jj}][\Phi] = [\Lambda_j] \end{cases} \tag{7}$$

where $[\Lambda_j] = diag[p_1^2, \cdots p_k^2, \cdots p_m^2]$ and $p_k(k = 1, 2, \cdots, m)$ is the natural frequency under the condition that the super-element has a fixed interface. $M$ is the degree of freedom inside the super-element. It can be obtained from Equation (7).

$$\begin{cases} \left[m_{jj}\right] = [\Phi]^{-T}[\Phi]^{-1} \\ \left[k_{jj}\right]^{-1} = [\Phi][\Lambda_j][\Phi]^T \\ \left[\Lambda_j\right]^{-1} = diag\left[\frac{1}{p_1^2}, \cdots, \frac{1}{p_r^2}, \cdots, \frac{1}{p_m^2}\right] \end{cases} \tag{8}$$

calculate the matrix in Equation (3) from the above equation $\left[k_{jj} - \omega^2 m_{jj}\right]^{-1}$.

$$\begin{aligned} & \left[k_{jj} - \omega^2 m_{jj}\right]^{-1} \\ &= \left[k_{jj} - \omega^2 m_{jj}k_{jj}^{-1}k_{jj}\right]^{-1} \\ &= [k_{jj}]^{-1}\left[I - \omega^2\Phi^{-T}\Phi^{-1}\Phi\Lambda_j\Phi^T\right]^{-1} \\ &= [k_{jj}]^{-1}\left[I - \omega^2\Phi^{-T}\Lambda_j\Phi^T\right]^{-1} \\ &= [k_{jj}]^{-1}[\Phi]^{-1}[T][\Phi]^{-T} \\ &= [k_{jj}]^{-1}[I + T_1] \end{aligned} \tag{9}$$

where

$$\begin{cases} [T] = diag\left[\frac{p_1^2}{p_1^2-\omega^2}, \cdots, \frac{p_k^2}{p_k^2-\omega^2}, \cdots, \frac{p_m^2}{p_m^2-\omega^2}\right] \\ [T_1] = [\Phi]^{-1}[\Lambda][\Phi]^T \\ [\Lambda] = diag\left[\frac{W^2}{p_1^2-\omega^2}, \cdots, \frac{W^2}{p_k^2-\omega^2}, \cdots, \frac{W^2}{p_m^2-\omega^2}\right] \end{cases}$$

in Equation (3), $[\beta]$ can be written as

$$[\beta] = -\left[k_{jj}\right]^{-1}[I + T_1]\left[K_{ji} - \omega^2 m_{ji}\right] \tag{10}$$

and substituting this equation into Equations (4) and (5) we obtain

$$\begin{cases} [\overline{m}] = [m_0] + 2\left[A\Lambda A^T\right] + \left[A\Lambda\Lambda A^T\right] \\ \left[\overline{k}\right] = [k_0] + \left[A\Lambda\Lambda_j\Lambda A^T\right] \end{cases} \tag{11}$$

where

$$\begin{cases} [m_0] = \left[ m_{ii} - m_{ij}k_{jj}^{-1}k_{ji} - k_{ij}k_{jj}^{-1}m_{ji} + k_{ij}k_{jj}^{-1}m_{jj}k_{jj}^{-1}k_{ji} \right] \\ [k_0] = \left[ k_{ii} = k_{ij}k_{jj}^{-1}k_{ji} \right] \\ [A] = \left[ m_{ii}\Phi - k_{ij}\Phi\Lambda_j^{-1} \right] \end{cases}$$

$$\left[ K_0 - \omega^2 M(\omega) \right]\{x_i\} = \{\overline{F}_i\} \tag{12}$$

where

$$\begin{cases} [K_0] = [k_0] \\ [M(\omega)] = \left[ m_0 + A\Lambda A^T \right] \end{cases}$$

The above derivation uses precise power reduction. Compared with static shrinkage, dynamic shrinkage introduces the inertia correction term $A\Lambda A^T$ on the basis of the static shrinkage term $[k_0]$ and $[m_0]$. The modified inertia term $[M(\omega)]$ is different from the static reduction value, while the elastic term is unchanged.

It should be noted that the above derivation does not introduce approximation. In practical applications, the higher order modes of the main modes of the fixed interface are generally omitted, and only some of the lower order modes are taken, thus greatly reducing the scale of analysis and calculation.

### 2.2. Substructure Modal Synthesis

The reduced super-element group is integrated into the motion equation of the whole system by using the conditions of interface displacement coordination and interface force balance.

$$\left[ \overline{K} \right]\{\overline{x}\} - \omega^2 \left[ \overline{M}(\omega) \right]\{\overline{x}\} = \{\overline{F}\} \tag{13}$$

The difference between this equation and the equations of motion of the whole system obtained by other substructure synthesis techniques is that the mass matrix is a function of frequency. Equation (13) corresponds to the nonlinear eigenvalue problem. This kind of eigenvalue problem can adopt the dichotomy method or other methods to solve the nonlinear eigenvalue problem.

## 3. Results and Discussion

### 3.1. Dynamic Frequency Calculation and Precision Check

Firstly, the analysis accuracy of the modal synthesis super-element method for prestressed and free interfacial substructures with fixed interfaces was verified. The dynamic frequency of the standard contact bladed disk system under the working speed was analyzed by using the cyclic symmetry analysis method and the modal synthesis super-element method of fixed interface prestressed and free interface substructures, respectively. In Table 1, the dimensionless dynamic frequency and relative error of the homophonic standard contact bladed disk system was calculated by two methods, and the working speeds are given. Figure 3 shows the dynamic frequency curve of the harmonized standard contact bladed disk system calculated by the two methods under the working speed. It can be seen that compared with the cyclic symmetry analysis method, the maximum relative error of the dimensionless dynamic frequency of the modal synthesis super-element method of the prestressed free interface substructure with a fixed interface is 5.68%. Since the number of modes intercepted by the substructures is the same, and the same finite element mesh model is used, the errors of the two methods at each frequency are relatively consistent.

**Table 1.** Accuracy verification of the modal synthesis super-element method for the prestressed and free interfacial substructures with fixed interfaces.

| Number of Knuckle Diameter | Cyclic Symmetric Structure Analysis Is Used to Harmonize the Dimensionless Dynamic Frequency of Standard Contact Bladed Disk System | Modal Synthesis Super-element Method of Fixed Interface Prestressed Free Interface Substructure Harmonizes the Dimensionless Dynamic Frequency of Standard Contact Bladed Disk System | Method Error |
|---|---|---|---|
| 0 | 1.0478 | 0.9889 | 5.62% |
| 1 | 1.0492 | 0.9903 | 5.61% |
| 2 | 1.0522 | 0.9936 | 5.57% |
| 3 | 1.0559 | 0.9975 | 5.54% |
| 4 | 1.0596 | 1.0011 | 5.52% |
| 5 | 1.0629 | 1.0042 | 5.52% |
| 6 | 1.0659 | 1.0069 | 5.54% |
| 7 | 1.0686 | 1.0091 | 5.56% |
| 8 | 1.0707 | 1.0109 | 5.59% |
| 9 | 1.0725 | 1.0122 | 5.61% |
| 10 | 1.0738 | 1.0133 | 5.63% |
| 11 | 1.0747 | 1.0141 | 5.64% |
| 12 | 1.0755 | 1.0146 | 5.65% |
| 13 | 1.0760 | 1.0151 | 5.66% |
| 14 | 1.0764 | 1.0154 | 5.67% |
| 15 | 1.0766 | 1.0156 | 5.67% |
| 16 | 1.0768 | 1.0157 | 5.67% |
| 17 | 1.0769 | 1.0158 | 5.68% |
| 18 | 1.0770 | 1.0159 | 5.68% |
| 19 | 1.0770 | 1.0159 | 5.68% |

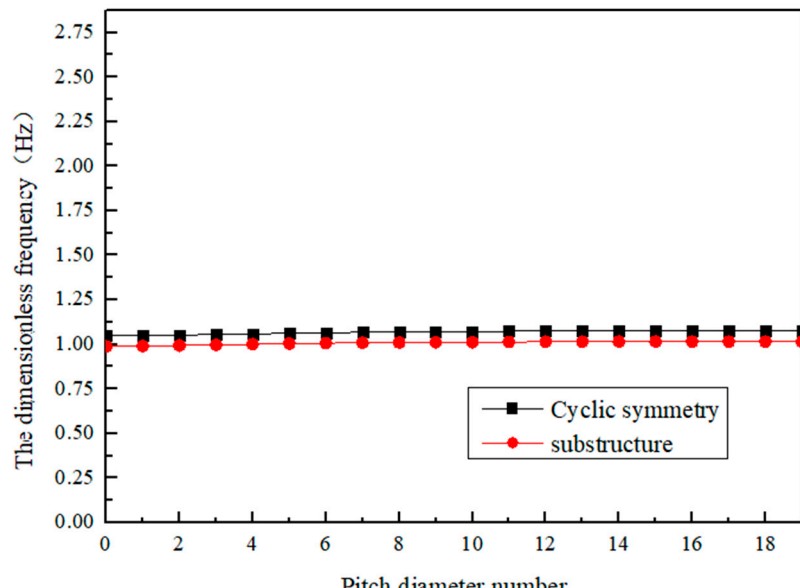

**Figure 3.** First-order dynamic frequency of the substructure method and cyclic commutation method.

## 3.2. Static Frequency Analysis of Blades

Since the natural vibration characteristics of the blades have a direct impact on the coupled vibration of the bladed disk system, the blades of the first and second stage of the bladed disk system were taken as the research objects, and the static frequency analysis of the two stages was carried out to obtain the inherent vibration characteristics. The three-dimensional solid model of the first- and second-stage bladed disk system and the finite element model of the two-stage blade after meshing are shown in Figures 4 and 5.

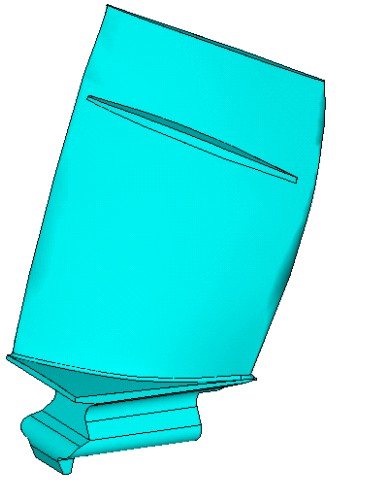

**Figure 4.** Single-blade model of Stage 1.

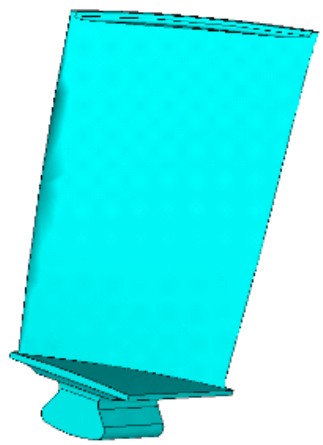
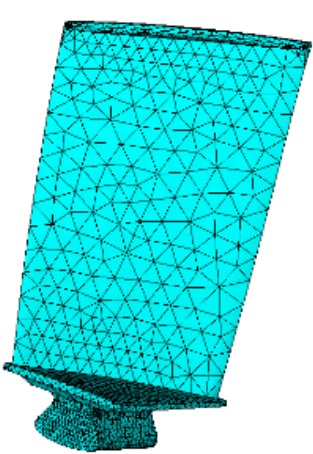

**Figure 5.** Single-blade model of Stage 2.

The first 10 order natural frequencies and mode shapes of the first- and second-stage blades were obtained by modal analysis after the tenon position of the first- and second-stage blades were all constrained.

The first 10 natural frequencies are shown in Table 2.

**Table 2.** Natural frequencies of the first 10 orders of the Stage 1 and Stage 2 blades.

| Stage 1 Blade | | Stage 2 Blade | |
|---|---|---|---|
| Order Number | Natural Frequency/Hz | Order Number | Natural Frequency/Hz |
| 1 | 602.0 | 1 | 632.0 |
| 2 | 1842.6 | 2 | 2273.2 |
| 3 | 2310.6 | 3 | 3043.1 |
| 4 | 4243.0 | 4 | 5590.8 |
| 5 | 4754.5 | 5 | 6616.9 |
| 6 | 5760.7 | 6 | 7898.7 |
| 7 | 6231.1 | 7 | 8803.8 |
| 8 | 6756.8 | 8 | 10,457.0 |
| 9 | 9294.0 | 9 | 12,714.0 |
| 10 | 9540.4 | 10 | 13,367.0 |

The first four natural frequencies and mode shapes are shown in Table 3:

**Table 3.** First four order natural frequencies and mode shapes of the Stage 1 and Stage 2 blades.

| Order Number | Stage 1 Blade | | Stage 2 Blade | |
| --- | --- | --- | --- | --- |
| | Natural Frequency/Hz | Vibration Mode | Natural Frequency/Hz | Vibration Mode |
| 1 | 602.0 | The first-order bending | 632.0 | The first-order bending |
| 2 | 1842.6 | The first-order distortion | 2273.2 | The first-order distortion |
| 3 | 2310.6 | The second-order bending | 3043.1 | The second-order bending |
| 4 | 4243.0 | The second-order distortion | 5590.8 | The second-order distortion |

Specific mode shapes are shown in Figures 6 and 7.

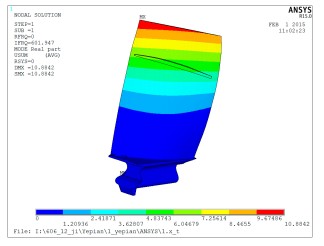 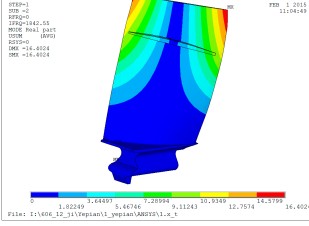 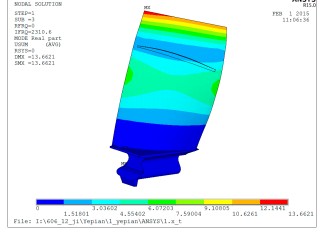 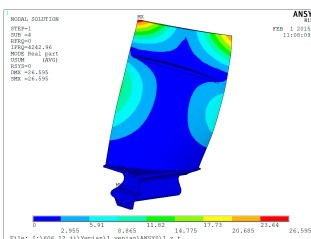

The first order     The second order     The third order     The fourth order

**Figure 6.** The first four modes of the first stage blade.

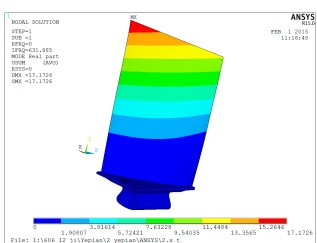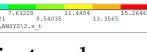 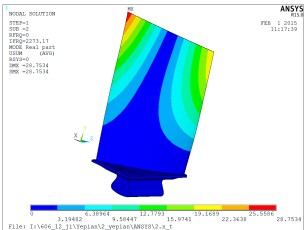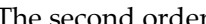 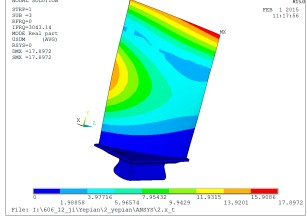

The first order     The second order     The third order     The fourth order

**Figure 7.** The first four modes of the second stage blade.

Through the analysis of the modes and vibration shapes of the first- and second-stage blades, it can be seen that the low-order mode shapes of the blades are bending and torsional vibration, and the frequency of the corresponding mode shapes of the second stage blades is slightly higher than that of the first-stage blades.

### 3.3. Modal Analysis of a Single-Stage Bladed Disk System

3.3.1. Modal Analysis of the First-Stage Bladed Disk System

The first-stage bladed disk system model was taken as the object of analysis, and its modal analysis was carried out. Figure 8 shows the three-dimensional solid model of the first-stage bladed disk system, and Figure 9 shows the finite element model of the first-stage bladed disk system. Through modal analysis, the first 150 order natural frequencies and mode shapes of the first-stage blading disk system were calculated and solved. The specific values of natural frequencies of each order are shown in Tables 4 and 5.

The first 150th order mode shapes of the bladed disk system of the first stage are shown in Table 5:

The mode diagram of the typical order of the first-stage bladed disk system is as follows (Figure 10):

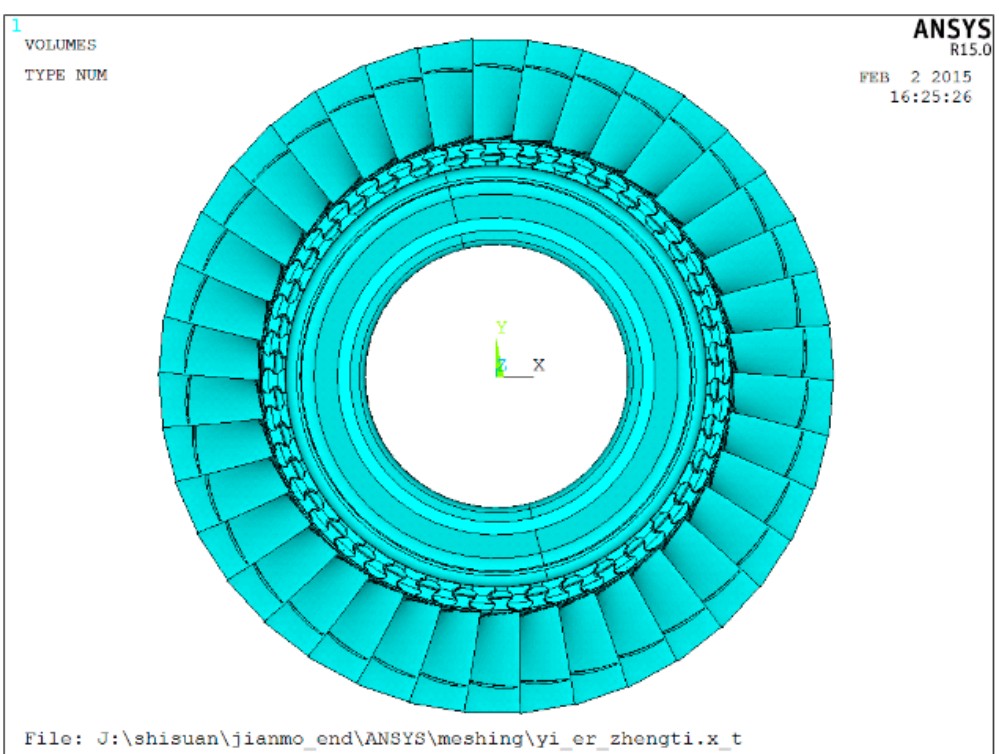

**Figure 8.** Three-dimensional solid model of the first-stage leaf disk.

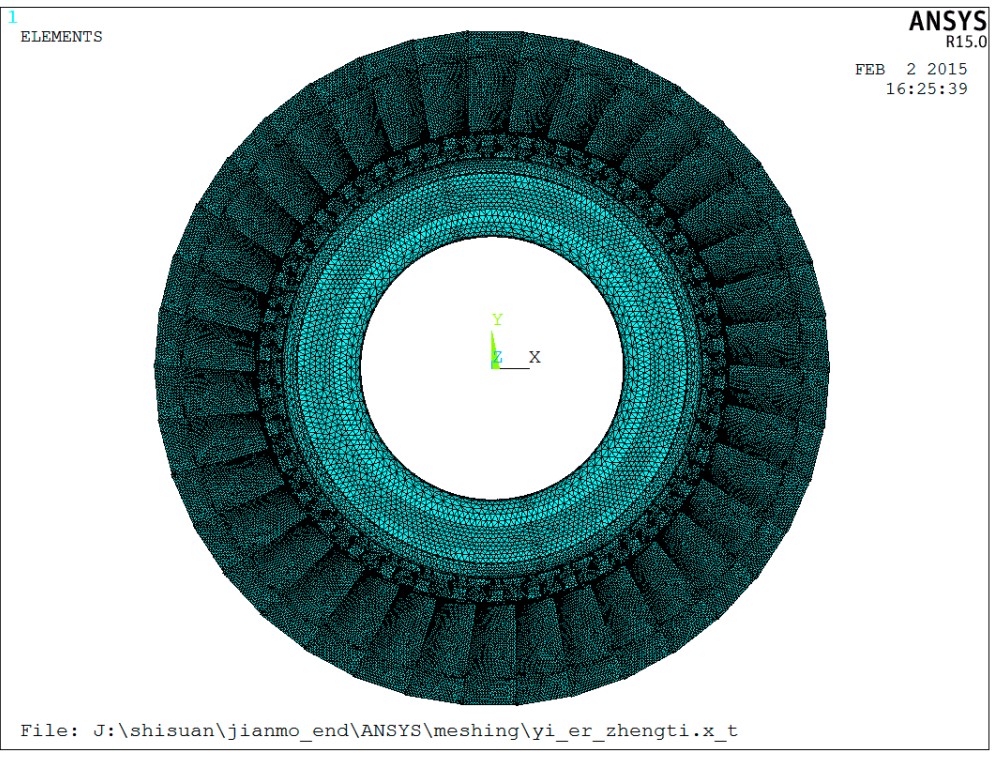

**Figure 9.** Finite element model of the first-stage bladed disk.

**Table 4.** Natural frequencies of the first 150 orders of the first-stage blading disk system.

| Order Number | Natural Frequency | Order Number | Natural Frequency | Order Number | Natural Frequency | Order Number | Natural Frequency | Order Number | Natural Frequency |
|---|---|---|---|---|---|---|---|---|---|
| 1 | 667.68 | 31 | 695.3 | 61 | 1813.10 | 91 | 1855.90 | 121 | 2189.90 |
| 2 | 668.27 | 32 | 695.33 | 62 | 1813.50 | 92 | 1870.10 | 122 | 2196.70 |
| 3 | 669.53 | 33 | 695.37 | 63 | 1814.10 | 93 | 1905.80 | 123 | 2198.40 |
| 4 | 671.51 | 34 | 695.40 | 64 | 1814.40 | 94 | 1914.30 | 124 | 2216.40 |
| 5 | 673.57 | 35 | 695.43 | 65 | 1814.70 | 95 | 1942.30 | 125 | 2218.50 |
| 6 | 677.21 | 36 | 695.45 | 66 | 1815.00 | 96 | 1948.30 | 126 | 2255.50 |
| 7 | 678.36 | 37 | 695.46 | 67 | 1815.20 | 97 | 1963.70 | 127 | 2255.80 |
| 8 | 682.41 | 38 | 695.48 | 68 | 1815.40 | 98 | 1968.80 | 128 | 2277.60 |
| 9 | 683.19 | 39 | 1064.10 | 69 | 1815.60 | 99 | 1976.30 | 129 | 2316.50 |
| 10 | 686.42 | 40 | 1208.30 | 70 | 1815.70 | 100 | 1980.80 | 130 | 2317.80 |
| 11 | 687.08 | 41 | 1217.00 | 71 | 1815.80 | 101 | 1984.20 | 131 | 2331.20 |
| 12 | 689.28 | 42 | 1317.70 | 72 | 1815.90 | 102 | 1988.00 | 132 | 2334.40 |
| 13 | 689.90 | 43 | 1331.70 | 73 | 1816.00 | 103 | 1989.40 | 133 | 2396.50 |
| 14 | 691.25 | 44 | 1455.40 | 74 | 1816.10 | 104 | 1992.50 | 134 | 2398.80 |
| 15 | 691.85 | 45 | 1456.30 | 75 | 1816.20 | 105 | 1993.20 | 135 | 2479.40 |
| 16 | 692.55 | 46 | 1460.20 | 76 | 1816.30 | 106 | 1995.30 | 136 | 2481.80 |
| 17 | 693.13 | 47 | 1573.40 | 77 | 1816.30 | 107 | 1996.00 | 137 | 2551.80 |
| 18 | 693.41 | 48 | 1584.50 | 78 | 1816.30 | 108 | 1997.20 | 138 | 2553.80 |
| 19 | 693.94 | 49 | 1687.90 | 79 | 1816.40 | 109 | 1997.90 | 139 | 2609.60 |
| 20 | 694.00 | 50 | 1693.00 | 80 | 1823.50 | 110 | 1998.60 | 140 | 2611.20 |
| 21 | 694.37 | 51 | 1703.60 | 81 | 1823.90 | 111 | 1999.20 | 141 | 2653.90 |
| 22 | 694.50 | 52 | 1767.90 | 82 | 1824.20 | 112 | 1999.70 | 142 | 2655.40 |
| 23 | 694.65 | 53 | 1774.70 | 83 | 1824.70 | 113 | 2000.20 | 143 | 2687.30 |
| 24 | 694.82 | 54 | 1784.70 | 84 | 1825.60 | 114 | 2000.70 | 144 | 2688.90 |
| 25 | 694.89 | 55 | 1797.20 | 85 | 1826.20 | 115 | 2001.00 | 145 | 2712.60 |
| 26 | 695.01 | 56 | 1806.20 | 86 | 1828.30 | 116 | 2001.20 | 146 | 2713.90 |
| 27 | 695.08 | 57 | 1806.40 | 87 | 1830.50 | 117 | 2001.40 | 147 | 2731.60 |
| 28 | 695.15 | 58 | 1810.20 | 88 | 1831.20 | 118 | 2143.70 | 148 | 2732.40 |
| 29 | 695.21 | 59 | 1811.20 | 89 | 1832.60 | 119 | 2159.80 | 149 | 2745.50 |
| 30 | 695.25 | 60 | 1812.30 | 90 | 1842.60 | 120 | 2186.90 | 150 | 2746.00 |

**Table 5.** The first 150 order natural frequencies and mode shapes of the first-stage bladed disk system.

| Order Number | Natural Frequency/Hz | Vibration Mode |
|---|---|---|
| 1–38 | 667.68–695.48 | Blade first-order bending vibration |
| 39–54 | 1064.10–1784.70 | The blade and the wheel vibrate together |
| 55–92 | 1797.20–1870.10 | First-order torsional vibration of blades |
| 93–117 | 1905.80–2001.40 | Blade bending-torsion coupled vibration |
| 118–150 | 2143.70–2746.00 | Second-order bending vibration of blade |

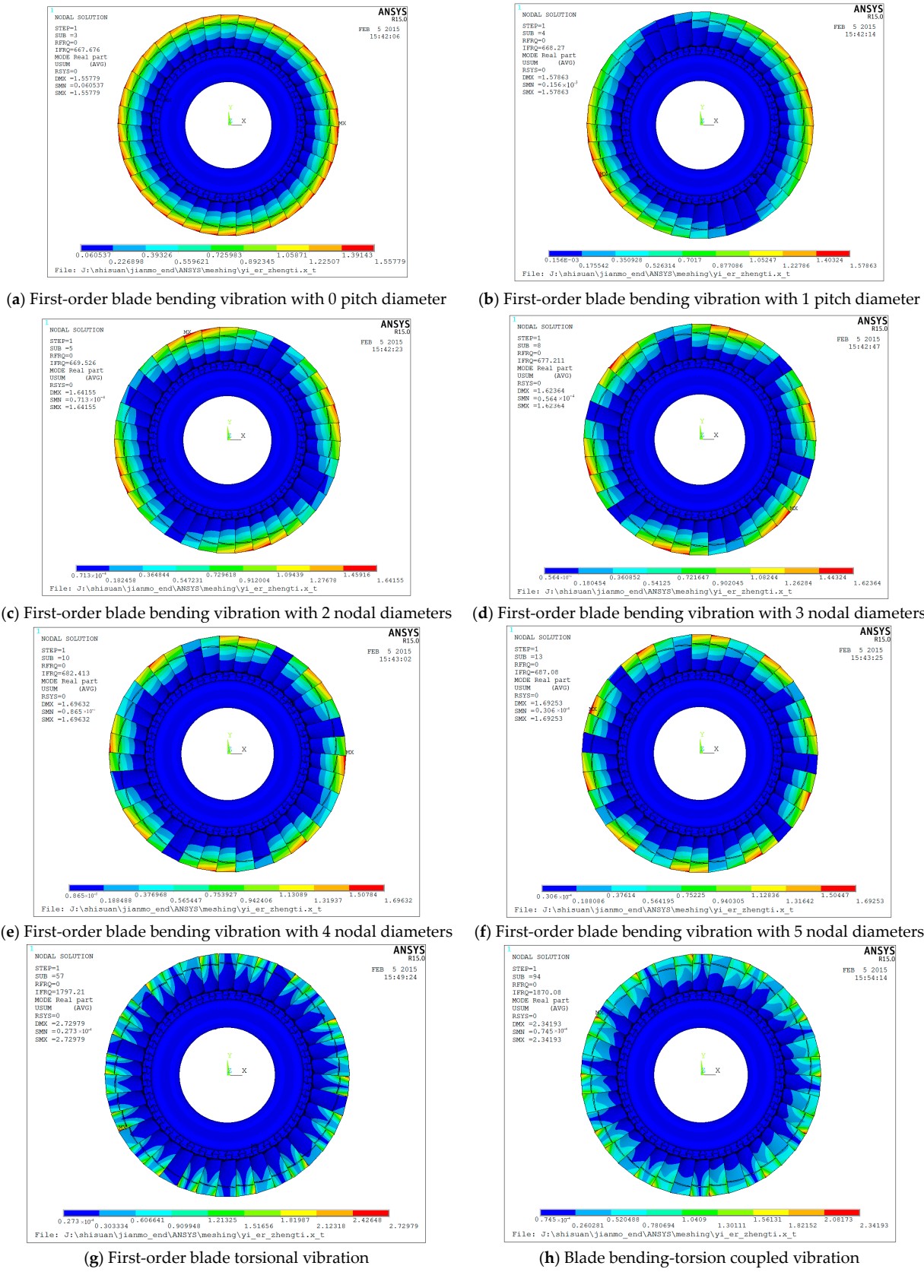

**Figure 10.** Mode pattern of a typical order of the first-stage bladed disk system.

Through analysis, it can be seen that the low order mode shape of the first-stage bladed disk system is the first order bending vibration of the blade according to the pitch

diameter. With the increase in mode order, the vibration of the disk is excited, and the coupled vibration of the blade and the disk appears. As the modal order continues to increase, the blade begins to transform from a bending vibration to twisting vibration. In addition, it can be found that due to the coupled action of the blade and the wheel, the first-order bending frequency of the blade is increased.

### 3.3.2. Modal Analysis of the Second-Stage Bladed Disk System

The second-stage bladed disk system model is taken as the analysis object, and its modal analysis is carried out. Figure 11 shows the three-dimensional solid model of the second-stage bladed disk system, and Figure 12 shows the finite element model of the second-stage bladed disk system. Through modal analysis, the first 150 order natural frequencies and mode shapes of the second-stage blading disk system were calculated and solved. The specific values of natural frequencies of each order are shown in Tables 6 and 7.

The first 150th order mode shapes of the second stage bladed disk system are shown in Table 7:

Figure 13 shows the mode diagram of the typical order of the second-stage bladed disk system.

According to the analysis, the vibration law of the second-stage bladed disk system is similar to that of the first-stage bladed disk system. With the increase in the modal order, the first-order bending vibration of the blade is presented, and then the vibration of the disk is excited, resulting in the coupled vibration of the blade and disk. As the modal order continues to increase, the blade begins to transform from a bending vibration to twisting vibration. At the same time, due to the coupling of the blade and the disk, the first-order bending frequency of the blade is increased.

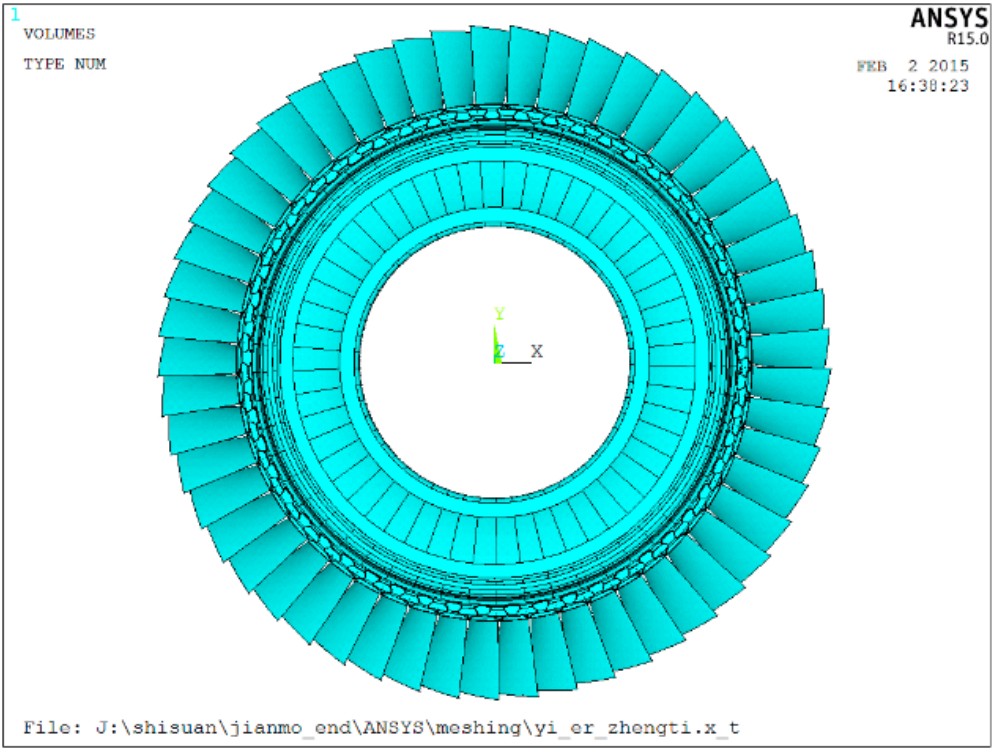

**Figure 11.** 3D solid model of the second-stage leaf disk.

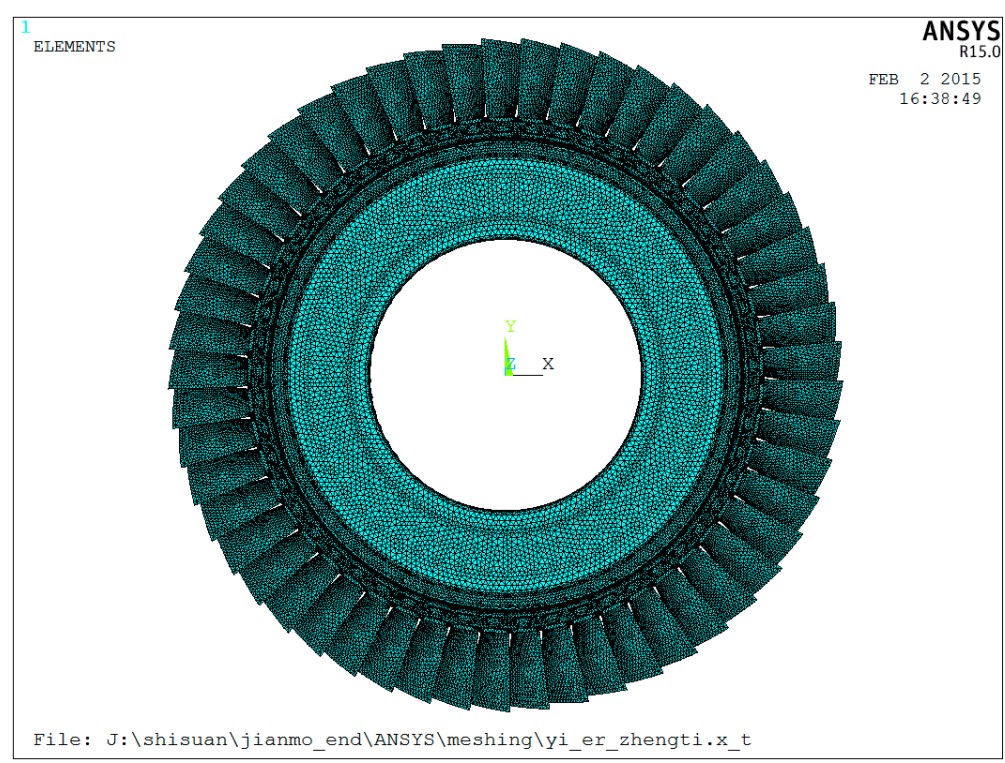

**Figure 12.** Finite element model of the second-stage bladed disk.

**Table 6.** Natural frequencies of the first 150 orders of the second-stage blading disk system.

| Order Number | Natural Frequency | Order Number | Natural Frequency | Order Number | Natural Frequency | Order Number | Natural Frequency | Order Number | Natural Frequency |
|---|---|---|---|---|---|---|---|---|---|
| 1 | 650.63 | 31 | 748.66 | 61 | 1561.70 | 91 | 2123.10 | 121 | 2249.80 |
| 2 | 706.22 | 32 | 748.73 | 62 | 1564.70 | 92 | 2123.80 | 122 | 2254.00 |
| 3 | 725.06 | 33 | 748.80 | 63 | 1610.80 | 93 | 2125.60 | 123 | 2255.50 |
| 4 | 725.58 | 34 | 748.85 | 64 | 1618.40 | 94 | 2126.30 | 124 | 2264.80 |
| 5 | 730.03 | 35 | 748.95 | 65 | 1657.50 | 95 | 2127.60 | 125 | 2267.70 |
| 6 | 733.46 | 36 | 748.98 | 66 | 1714.00 | 96 | 2128.30 | 126 | 2290.40 |
| 7 | 734.18 | 37 | 749.00 | 67 | 1718.80 | 97 | 2129.30 | 127 | 2293.50 |
| 8 | 737.91 | 38 | 749.07 | 68 | 1833.40 | 98 | 2129.90 | 128 | 2324.40 |
| 9 | 738.53 | 39 | 749.10 | 69 | 1838.50 | 99 | 2130.70 | 129 | 2327.60 |
| 10 | 740.91 | 40 | 749.12 | 70 | 1850.40 | 100 | 2131.20 | 130 | 2361.80 |
| 11 | 741.53 | 41 | 749.17 | 71 | 1930.90 | 101 | 2131.90 | 131 | 2364.90 |
| 12 | 742.98 | 42 | 749.19 | 72 | 1934.90 | 102 | 2132.20 | 132 | 2396.30 |
| 13 | 743.63 | 43 | 749.24 | 73 | 1971.90 | 103 | 2132.80 | 133 | 2399.50 |
| 14 | 744.48 | 44 | 749.26 | 74 | 2001.70 | 104 | 2133.10 | 134 | 2425.20 |
| 15 | 745.14 | 45 | 749.28 | 75 | 2004.30 | 105 | 2133.50 | 135 | 2428.40 |
| 16 | 745.57 | 46 | 749.29 | 76 | 2013.30 | 106 | 2133.70 | 136 | 2448.10 |
| 17 | 746.21 | 47 | 749.33 | 77 | 2047.30 | 107 | 2134.00 | 137 | 2451.30 |
| 18 | 746.35 | 48 | 749.37 | 78 | 2048.40 | 108 | 2134.30 | 138 | 2466.00 |
| 19 | 746.92 | 49 | 749.40 | 79 | 2075.10 | 109 | 2134.70 | 139 | 2469.10 |
| 20 | 746.99 | 50 | 749.42 | 80 | 2075.30 | 110 | 2135.00 | 140 | 2480.10 |

**Table 6.** *Cont.*

| Order Number | Natural Frequency | Order Number | Natural Frequency | Order Number | Natural Frequency | Order Number | Natural Frequency | Order Number | Natural Frequency |
|---|---|---|---|---|---|---|---|---|---|
| 21 | 747.39 | 51 | 749.43 | 81 | 2091.90 | 111 | 2135.00 | 141 | 2483.00 |
| 22 | 747.52 | 52 | 749.46 | 82 | 2092.40 | 112 | 2135.40 | 142 | 2491.20 |
| 23 | 747.75 | 53 | 749.50 | 83 | 2102.90 | 113 | 2135.40 | 143 | 2493.90 |
| 24 | 747.92 | 54 | 797.09 | 84 | 2103.50 | 114 | 2135.50 | 144 | 2500.10 |
| 25 | 748.02 | 55 | 835.36 | 85 | 2110.40 | 115 | 2230.70 | 145 | 2502.50 |
| 26 | 748.21 | 56 | 999.34 | 86 | 2111.20 | 116 | 2232.80 | 146 | 2507.30 |
| 27 | 748.27 | 57 | 1002.60 | 87 | 2115.80 | 117 | 2233.00 | 147 | 2509.30 |
| 28 | 748.42 | 58 | 1305.60 | 88 | 2116.60 | 118 | 2235.90 | 148 | 2513.10 |
| 29 | 748.46 | 59 | 1306.80 | 89 | 2119.90 | 119 | 2238.20 | 149 | 2514.90 |
| 30 | 748.55 | 60 | 1500.20 | 90 | 2120.70 | 120 | 2247.20 | 150 | 2517.80 |

**Table 7.** First 150 order natural frequencies and mode shapes of stage 2 blading disk systems.

| Order Number | Natural Frequency/Hz | Vibration Mode |
|---|---|---|
| 1 | 650.63 | The vibration of the blade and the wheel is coupled with 0 pitch diameter |
| 2–53 | 706.22–749.50 | Blade first-order bending vibration |
| 54–78 | 797.09–2048.40 | The blade and the wheel vibrate together |
| 79–131 | 2075.10–2364.90 | First-order torsional vibration of blades |
| 132–150 | 2396.30–2517.80 | Blade bending-torsion coupled vibration |

### 3.4. Modal Analysis of the Two-Stage Bladed Disk Coupled System

For the interstage coupled vibration analysis of multi-stage bladed disk system, the two-stage bladed disk system composed of the first- and second-stage bladed disk systems is selected firstly, and the overall model of the two-stage bladed disk system is taken as the analysis object. Figure 14a shows the overall three-dimensional solid model of the two-stage bladed disk system. Figure 14b shows the overall finite element model of a two-stage bladed disk system with meshing and boundary conditions considered. Through modal analysis, the first 195th order natural frequencies and vibration shapes of the two-stage bladed disk system were calculated and solved, as shown in Tables 8 and 9.

**Table 8.** First 195 order natural frequencies of a two-stage bladed disk coupled system.

| Order Number | Natural Frequency | Order Number | Natural Frequency | Order Number | Natural Frequency | Order Number | Natural Frequency | Order Number | Natural Frequency |
|---|---|---|---|---|---|---|---|---|---|
| 1 | 656.78 | 40 | 720.99 | 79 | 748.96 | 118 | 1763.90 | 157 | 1852.60 |
| 2 | 664.55 | 41 | 722.37 | 80 | 748.98 | 119 | 1766.50 | 158 | 1866.90 |
| 3 | 666.86 | 42 | 728.61 | 81 | 749.03 | 120 | 1781.10 | 159 | 1870.70 |
| 4 | 666.99 | 43 | 729.35 | 82 | 749.05 | 121 | 1795.60 | 160 | 1874.00 |
| 5 | 668.81 | 44 | 735.55 | 83 | 749.07 | 122 | 1805.10 | 161 | 1875.30 |
| 6 | 674.13 | 45 | 736.26 | 84 | 749.09 | 123 | 1805.60 | 162 | 1894.80 |
| 7 | 675.66 | 46 | 739.31 | 85 | 749.12 | 124 | 1809.70 | 163 | 1900.60 |
| 8 | 676.78 | 47 | 739.91 | 86 | 749.17 | 125 | 1810.60 | 164 | 1909.50 |
| 9 | 681.73 | 48 | 741.74 | 87 | 749.2 | 126 | 1811.90 | 165 | 1937.20 |
| 10 | 682.5 | 49 | 742.33 | 88 | 749.21 | 127 | 1812.70 | 166 | 1937.30 |
| 11 | 686.01 | 50 | 743.44 | 89 | 749.23 | 128 | 1813.20 | 167 | 1943.40 |

**Table 8.** *Cont.*

| Order Number | Natural Frequency | Order Number | Natural Frequency | Order Number | Natural Frequency | Order Number | Natural Frequency | Order Number | Natural Frequency |
|---|---|---|---|---|---|---|---|---|---|
| 12 | 686.67 | 51 | 744.06 | 90 | 749.25 | 129 | 1813.80 | 168 | 1952.80 |
| 13 | 688.97 | 52 | 744.68 | 91 | 749.30 | 130 | 1814.10 | 169 | 1955.20 |
| 14 | 689.58 | 53 | 745.31 | 92 | 778.35 | 131 | 1814.40 | 170 | 1959.10 |
| 15 | 690.95 | 54 | 745.60 | 93 | 1025.80 | 132 | 1814.70 | 171 | 1962.10 |
| 16 | 691.55 | 55 | 746.21 | 94 | 1026.20 | 133 | 1814.90 | 172 | 1964.10 |
| 17 | 692.26 | 56 | 746.29 | 95 | 1037.40 | 134 | 1815.20 | 173 | 1967.90 |
| 18 | 692.84 | 57 | 746.80 | 96 | 1195.50 | 135 | 1815.30 | 174 | 1972.00 |
| 19 | 693.13 | 58 | 746.89 | 97 | 1199.00 | 136 | 1815.40 | 175 | 1976.40 |
| 20 | 693.66 | 59 | 747.23 | 98 | 1284.50 | 137 | 1815.60 | 176 | 1980.00 |
| 21 | 693.73 | 60 | 747.38 | 99 | 1292.50 | 138 | 1815.70 | 177 | 1983.90 |
| 22 | 694.09 | 61 | 747.57 | 100 | 1358.10 | 139 | 1815.80 | 178 | 1985.40 |
| 23 | 694.23 | 62 | 747.75 | 101 | 1359.40 | 140 | 1815.90 | 179 | 1988.50 |
| 24 | 694.39 | 63 | 747.83 | 102 | 1420.90 | 141 | 1816.00 | 180 | 1989.30 |
| 25 | 694.56 | 64 | 748.02 | 103 | 1426.50 | 142 | 1816.00 | 181 | 1991.50 |
| 26 | 694.63 | 65 | 748.08 | 104 | 1471.40 | 143 | 1816.10 | 182 | 1992.20 |
| 27 | 694.76 | 66 | 748.22 | 105 | 1484.90 | 144 | 1816.10 | 183 | 1993.50 |
| 28 | 694.82 | 67 | 748.26 | 106 | 1548.50 | 145 | 1816.20 | 184 | 1994.20 |
| 29 | 694.90 | 68 | 748.35 | 107 | 1555.20 | 146 | 1819.50 | 185 | 1994.90 |
| 30 | 694.95 | 69 | 748.46 | 108 | 1566.60 | 147 | 1822.60 | 186 | 1995.50 |
| 31 | 695.00 | 70 | 748.52 | 109 | 1585.50 | 148 | 1823.10 | 187 | 1996.20 |
| 32 | 695.04 | 71 | 748.59 | 110 | 1604.20 | 149 | 1824.70 | 188 | 1996.60 |
| 33 | 695.08 | 72 | 748.65 | 111 | 1615.90 | 150 | 1825.20 | 189 | 1997.20 |
| 34 | 695.13 | 73 | 748.74 | 112 | 1618.00 | 151 | 1825.30 | 190 | 1997.50 |
| 35 | 695.15 | 74 | 748.78 | 113 | 1679.80 | 152 | 1826.20 | 191 | 1997.70 |
| 36 | 695.19 | 75 | 748.79 | 114 | 1695.50 | 153 | 1827.50 | 192 | 1997.90 |
| 37 | 695.21 | 76 | 748.86 | 115 | 1700.80 | 154 | 1830.10 | 193 | 2011.00 |
| 38 | 695.22 | 77 | 748.90 | 116 | 1730.40 | 155 | 1831.90 | 194 | 2012.60 |
| 39 | 695.23 | 78 | 748.92 | 117 | 1763.20 | 156 | 1841.20 | 195 | 2049.70 |

**Table 9.** First 160th order natural frequencies and mode shapes of the two-stage bladed disk coupled system.

| Order Number | Natural Frequency/Hz | VIBRATION MODE |
|---|---|---|
| 1 | 656.78 | Coupled vibration of the two-stage blade and wheel |
| 2–5 | 664.55–668.810 | First-order bending vibration of the one-stage blade |
| 6 | 674.13 | Coupled vibration of the two-stage blade and wheel |
| 7–39 | 675.66–695.23 | First-order bending vibration of the one-stage blade |
| 40–91 | 720.99–749.30 | First-order bending vibration of the two-stage blades |
| 92 | 778.35 | Zero pitch diameter vibration of the stage two blades and disks |
| 93–94 | 1025.80–1026.20 | One-pitch vibration of the stage two blades and disks |
| 95 | 1037.40 | Zero pitch diameter vibration of the stage one blade and wheel |
| 96–97 | 1195.50–1199.00 | One-pitch vibration of the stage one blades and disks |
| 98–99 | 1284.50–1292.50 | Two-pitch vibration of the stage one blades and disks |
| 100–101 | 1358.10–1359.40 | Two-pitch diameter vibration of the two stage blades and disks |
| 102–103 | 1420.90–1426.50 | Three-pitch vibration of the stage one blades and disks |
| 104–106 | 1471.40–1548.50 | Coupled vibration of the two-stage blade and wheel |

**Table 9.** *Cont.*

| Order Number | Natural Frequency/Hz | VIBRATION MODE |
|---|---|---|
| 107–108 | 1555.20–1566.60 | Four-pitch vibration of the stage one blades and disks |
| 109–110 | 1585.50–1604.20 | Two-pitch diameter vibration of the two-stage blades and disks |
| 111–112 | 1615.90–1618.00 | Three-section diameter vibration of the stage two blades and disks |
| 113–114 | 1679.80–1695.50 | Five section diameter vibration of the stage one blade and wheel |
| 115–116 | 1700.80–1730.40 | Coupled vibration of the two-stage blade and wheel |
| 117 | 1763.20 | Four-pitch vibration of the stage two blades and disks |
| 118 | 1763.90 | Bending-torsion coupled vibration of the stage one blade |
| 119 | 1766.50 | Four-pitch vibration of the stage two blades and disks |
| 120–158 | 1781.10–1866.90 | First order torsional vibration of the one-stage blade |
| 159–160 | 1870.70–1874.00 | Five section diameter vibration of the stage two blades and wheel |

The first 160th order mode shapes of the two-stage bladed disk coupled system are shown in Table 9:

The mode diagram of the typical order of the two-stage bladed disk coupled system is as follows:

(1) As shown in Figures 15–19, coupled mode shape when the first-stage bladed disk vibration is dominant.

(2) As shown in Figures 20–24, coupled mode shapes of the second-stage bladed disk when the vibration is dominant.

(3) As shown in Figures 25–29, coupled mode of two stages of the bladed disk with the same vibration amplitude.

According to the modal and mode shape analysis of the two-stage bladed disk coupled system, in the low-order mode, the first-order bending vibration of the first-order blade and the second-order blade is firstly manifested. With the increase in the modal order, the coupled vibration of the blade and the wheel, the twisted vibration of the blade, and the coupled vibration between the two stages of the bladed disk appear. In addition, it can be found that the vibration of the two-stage bladed disk will appear at the same frequency with the same pitch diameter. However, in most two-stage coupled bladed disks, the mode pattern cannot be seen directly. This is because the vibration of the first-stage bladed disk is dominant, while the vibration of the other stage is relatively small.

For the order with obvious inter-stage coupled vibration of the two-stage bladed disk system, the maximum vibration displacements of the first-stage bladed disk system and the second-stage bladed disk system are shown in Table 10. Figures 30–35 show the inter-stage coupled modes.

**Table 10.** Maximum displacement and ratio of the vibration of the Stage 1 and Stage 2 bladed disk systems.

| Order Number | Natural Frequency/Hz | Level One Blade Disk/mm | Level Two Blade Disk/mm | Ratio |
|---|---|---|---|---|
| 99 | 1292.5 | 1.0060 | 0.2319 | 4.34 |
| 104 | 1471.4 | 0.3266 | 0.1907 | 1.71 |
| 105 | 1484.9 | 0.4579 | 1.0943 | 2.39 |
| 106 | 1548.5 | 0.5240 | 0.9044 | 1.73 |
| 115 | 1700.8 | 0.5405 | 0.0969 | 5.58 |
| 116 | 1730.4 | 0.4907 | 0.5007 | 1.00 |

Note: the ratio is the ratio of the larger value to the smaller value in the maximum displacements of Stage 1 and Stage 2 bladed disk systems.

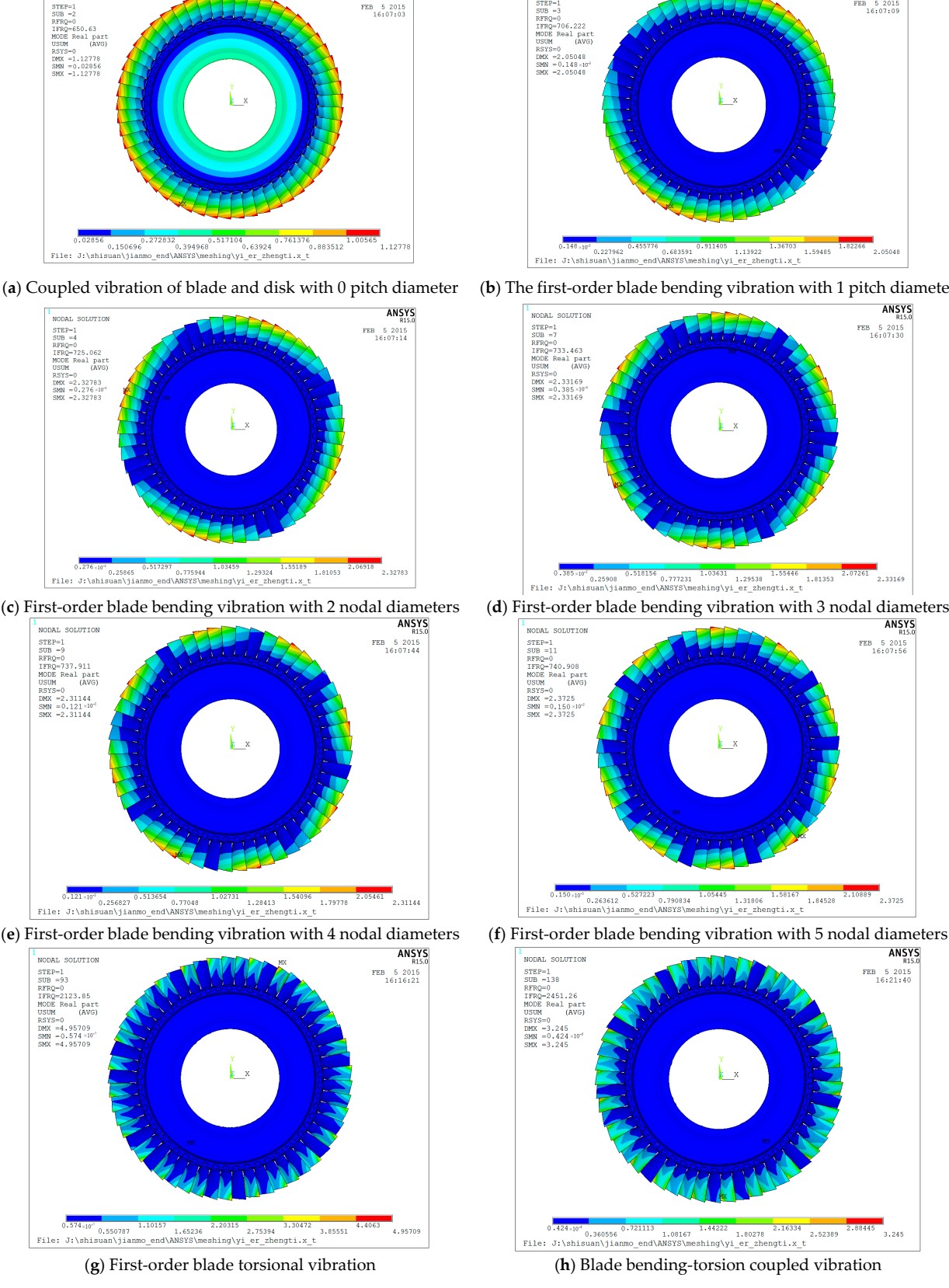

(**a**) Coupled vibration of blade and disk with 0 pitch diameter

(**b**) The first-order blade bending vibration with 1 pitch diameter

(**c**) First-order blade bending vibration with 2 nodal diameters

(**d**) First-order blade bending vibration with 3 nodal diameters

(**e**) First-order blade bending vibration with 4 nodal diameters

(**f**) First-order blade bending vibration with 5 nodal diameters

(**g**) First-order blade torsional vibration

(**h**) Blade bending-torsion coupled vibration

**Figure 13.** Mode pattern of a typical order of the second-stage bladed disk system.

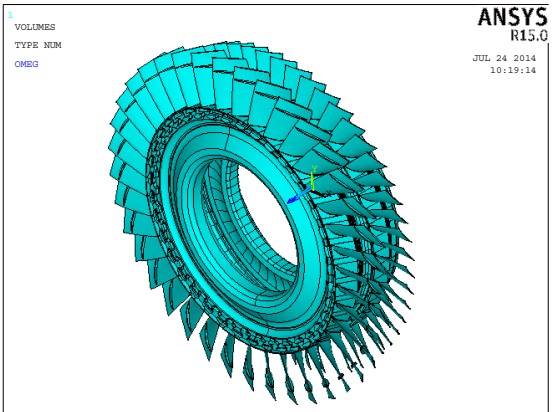

(**a**) 3D solid model of a two-stage bladed disk system

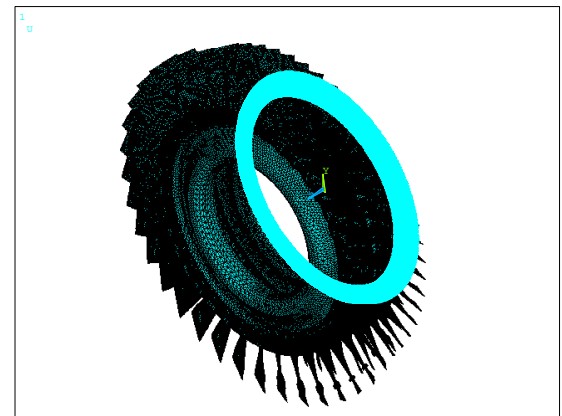

(**b**) Finite element model of a two-stage bladed disk system

**Figure 14.** Two-stage bladed disk system model.

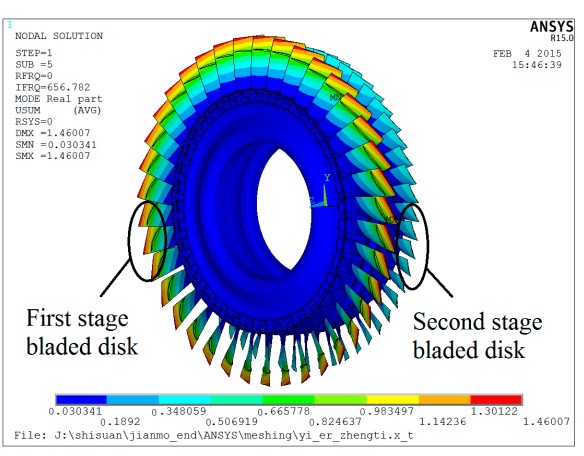

(**a**) Two-stage coupled bladed disk

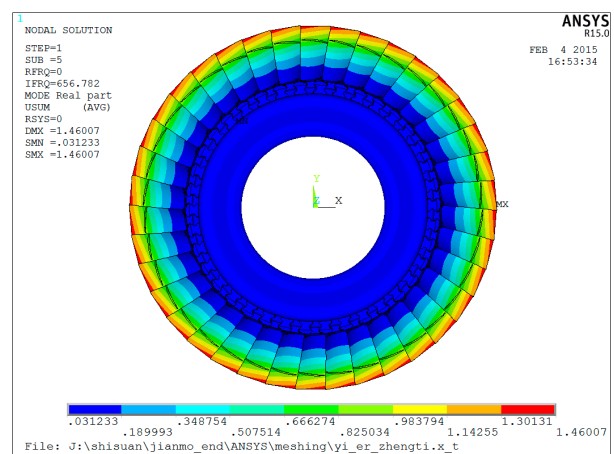

(**b**) First-stage bladed disk

**Figure 15.** One-pitch coupled vibration of the blade and disk.

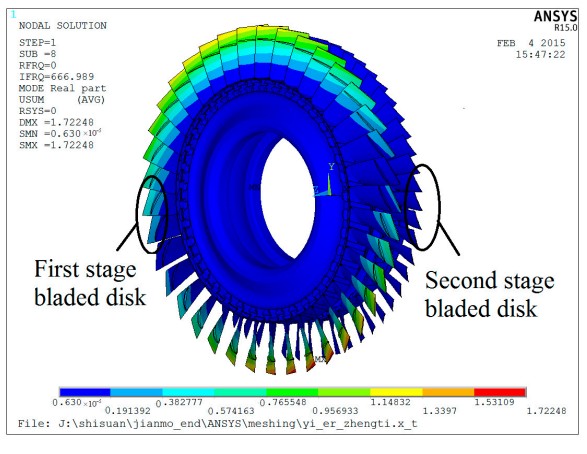

(**a**) Two-stage coupled bladed disk

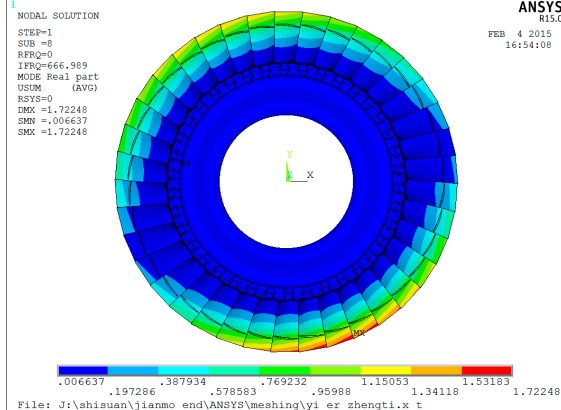

(**b**) First-stage bladed disk

**Figure 16.** One-pitch coupled vibration of the blade and disk.

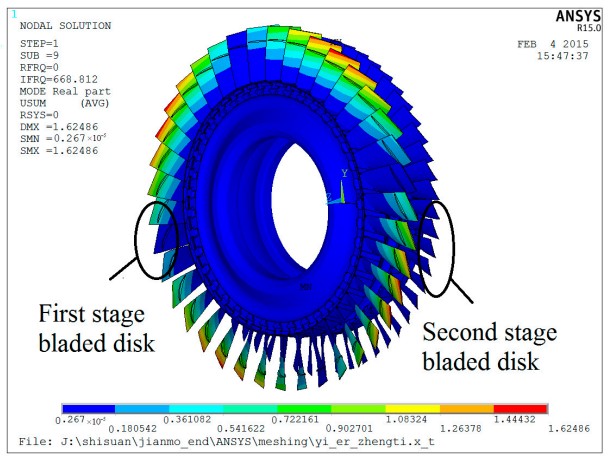 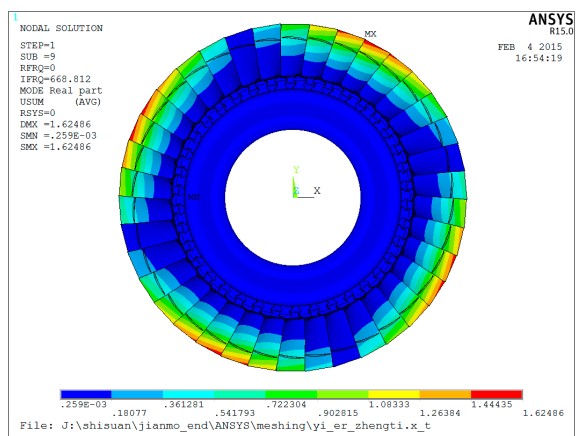

(**a**) Two-stage coupled bladed disk      (**b**) First-stage bladed disk

**Figure 17.** Two-pitch coupled vibration of the blade and disk.

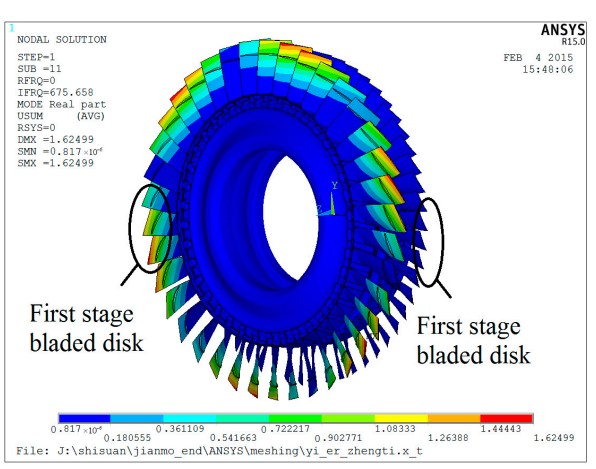 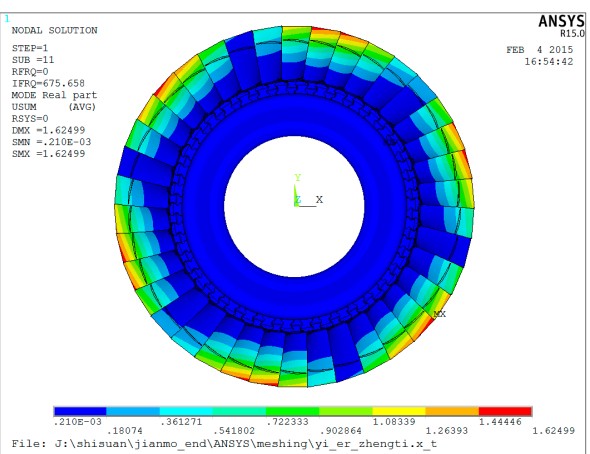

(**a**) Two-stage coupled bladed disk      (**b**) First-stage bladed disk

**Figure 18.** Coupled vibration of the blade and disk in three segments.

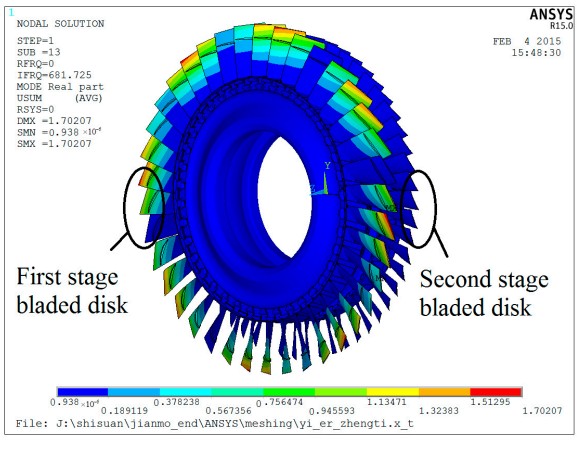 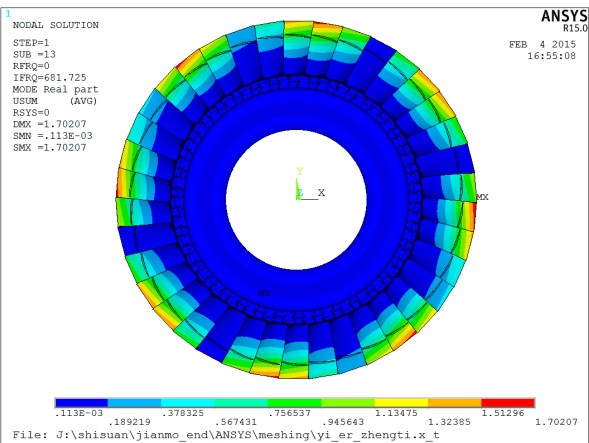

(**a**) Two-stage coupled bladed disk      (**b**) First-stage bladed disk

**Figure 19.** Four-pitch coupled vibration of the blade and disk.

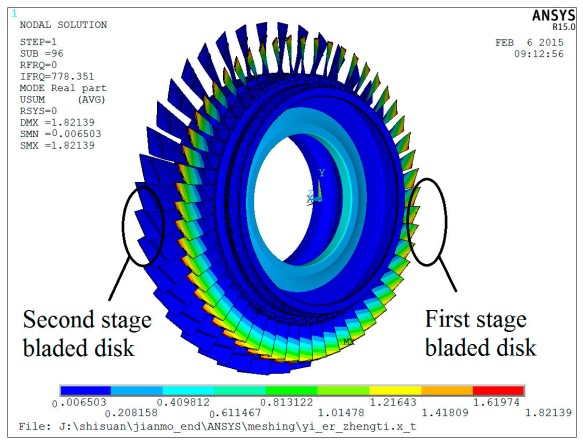 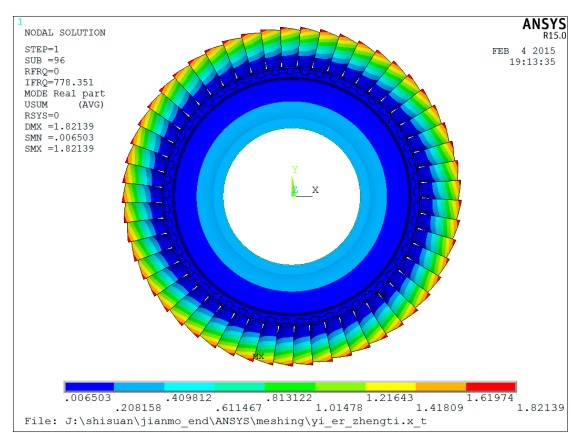

(**a**) Two-stage coupled bladed disk   (**b**) Second-stage bladed disk

**Figure 20.** Coupled vibration of the blade and disk with a 0 pitch diameter.

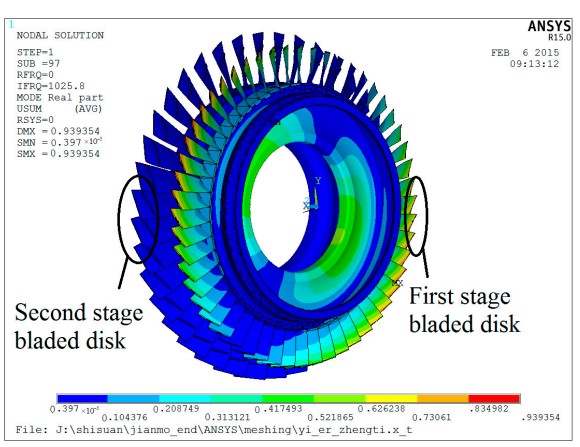 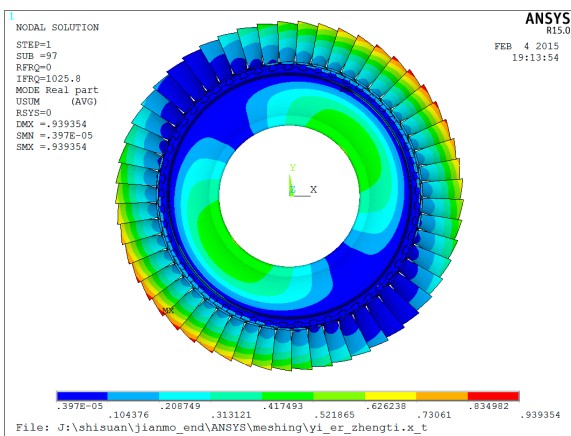

(**a**) Two-stage coupled bladed disk   (**b**) Second stage bladed disk

**Figure 21.** One-pitch coupled vibration of the blade and disk.

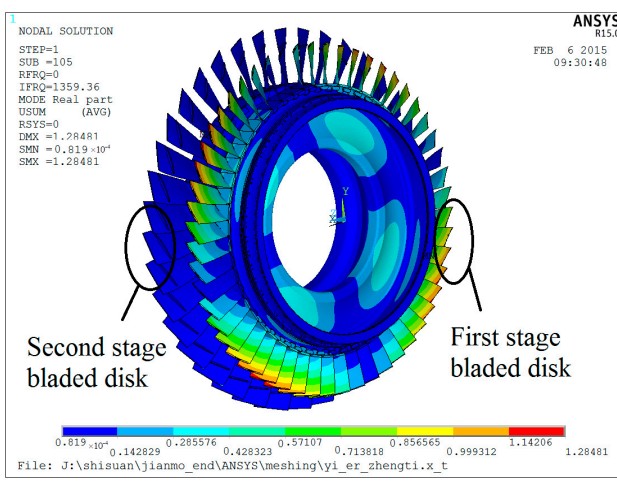 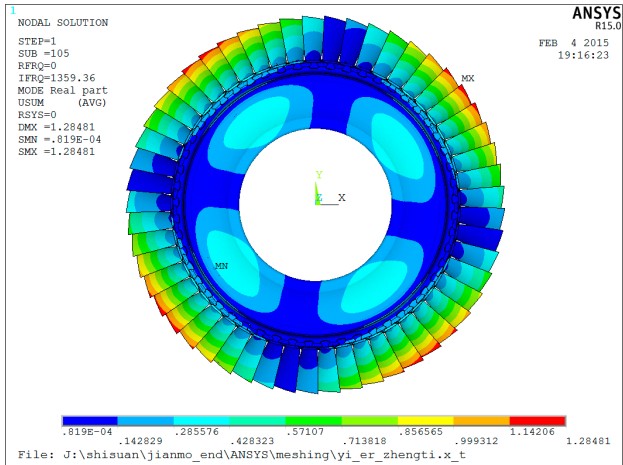

(**a**) Two-stage coupled bladed disk   (**b**) Second-stage bladed disk

**Figure 22.** Coupled vibration of the blade and disk with a two-pitch diameter.

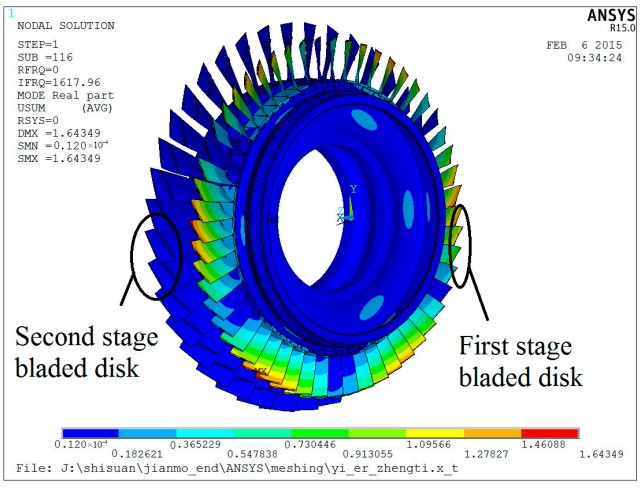

(**a**) Two-stage coupled bladed disk

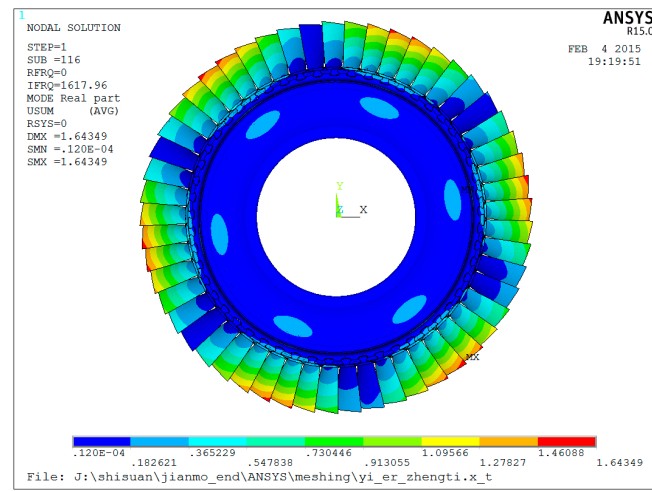

(**b**) Second-stage bladed disk

**Figure 23.** Coupled vibration of the blade and disk in three segments.

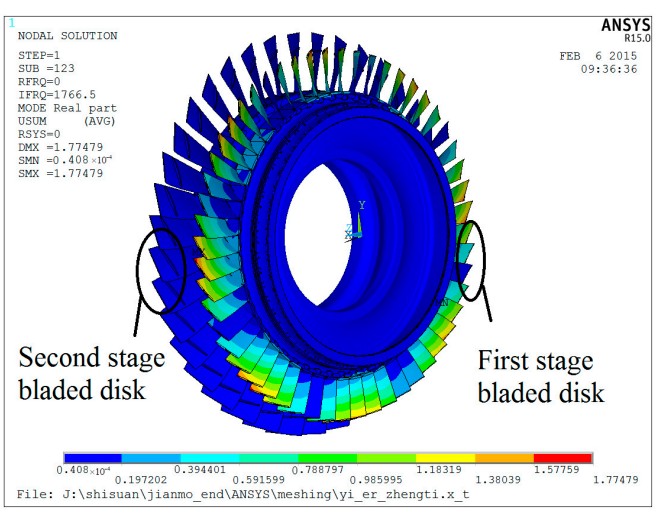

(**a**) Two-stage coupled bladed disk

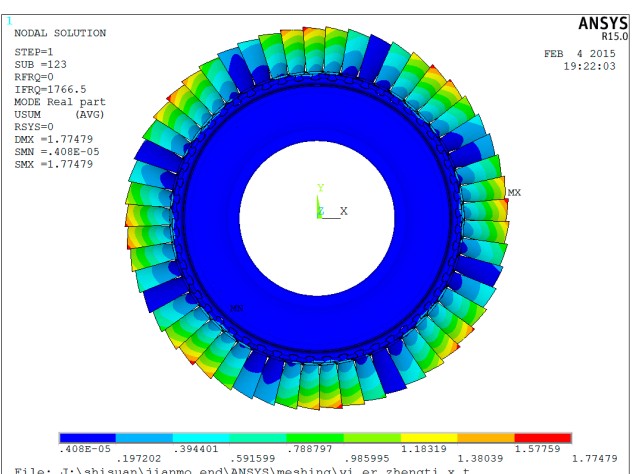

(**b**) Second-stage bladed disk

**Figure 24.** Four-pitch coupled vibration of the blade and disk.

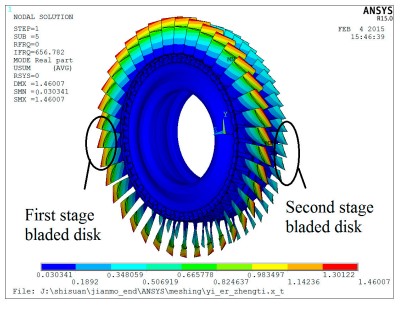

(**a**) Two-stage coupled bladed disk

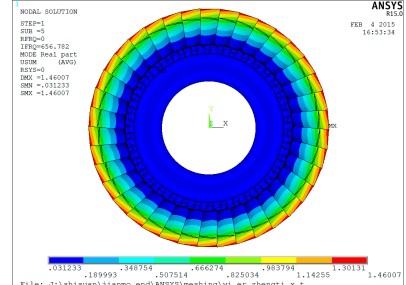

(**b**) First-stage bladed disk

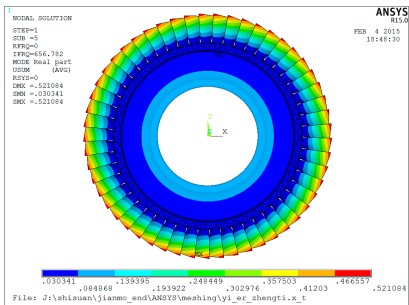

(**c**) Second-stage bladed disk

**Figure 25.** Coupled vibration at a natural frequency of 656.78 Hz.

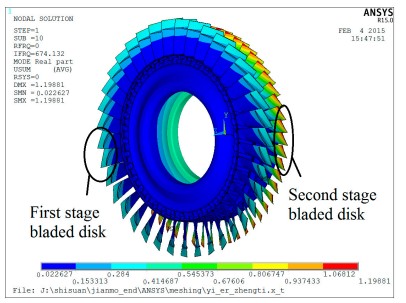 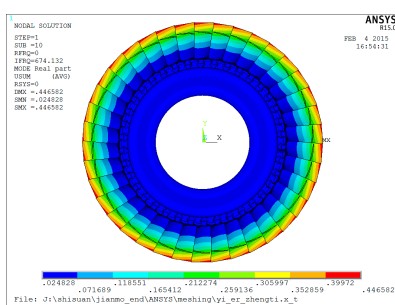 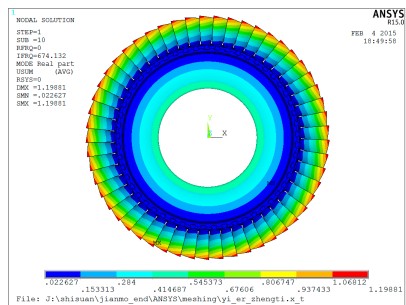

(**a**) Two-stage coupled bladed disk    (**b**) First-stage bladed disk    (**c**) Second-stage bladed disk

**Figure 26.** Coupled vibration at a 674.13 Hz natural frequency.

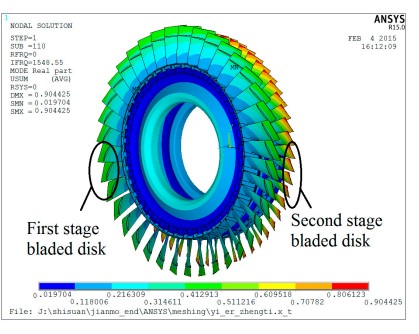 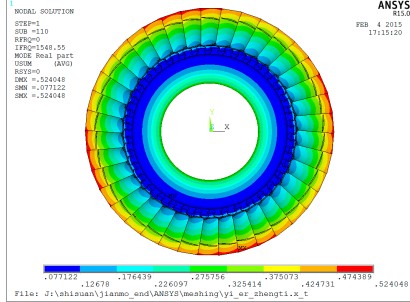 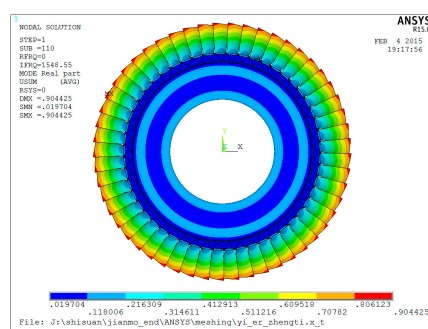

(**a**) Two-stage coupled bladed disk    (**b**) First-stage bladed disk    (**c**) Second-stage bladed disk

**Figure 27.** Coupled vibration at a natural frequency of 1548.55 Hz.

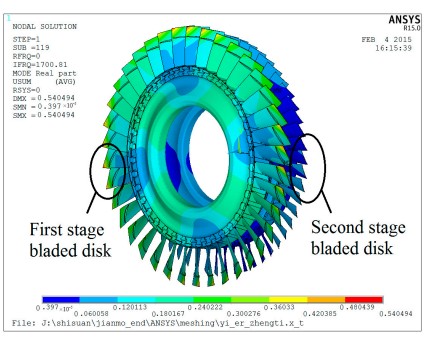 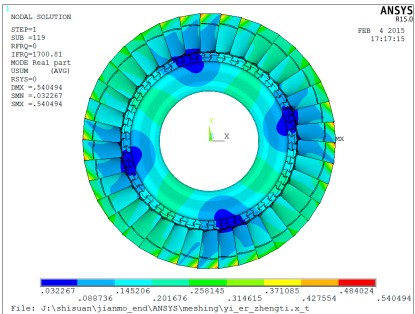 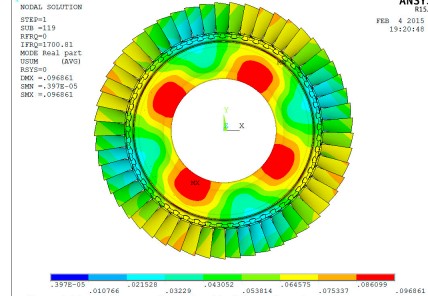

(**a**) Two-stage coupled bladed disk    (**b**) First-stage bladed disk    (**c**) Second-stage bladed disk

**Figure 28.** Coupled vibration at a natural frequency of 1700.81 Hz.

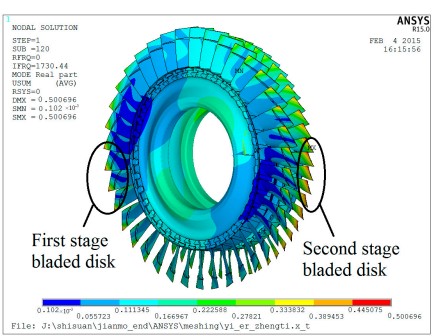

(**a**) Two-stage coupled bladed disk    (**b**) First-stage bladed disk    (**c**) Second-stage bladed disk

**Figure 29.** Coupled vibration at a natural frequency of 1730.44 Hz.

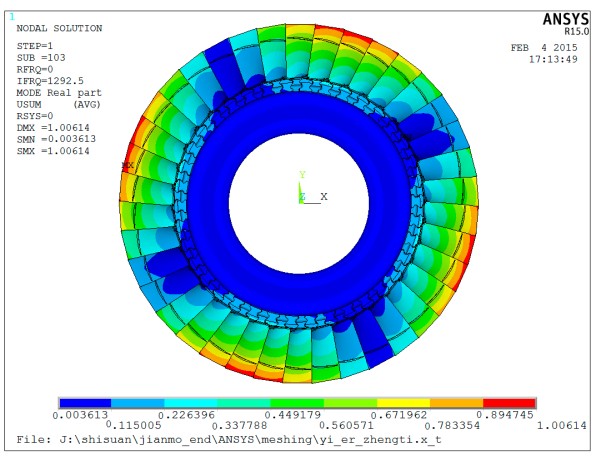

(**a**) Grade one leaf disk

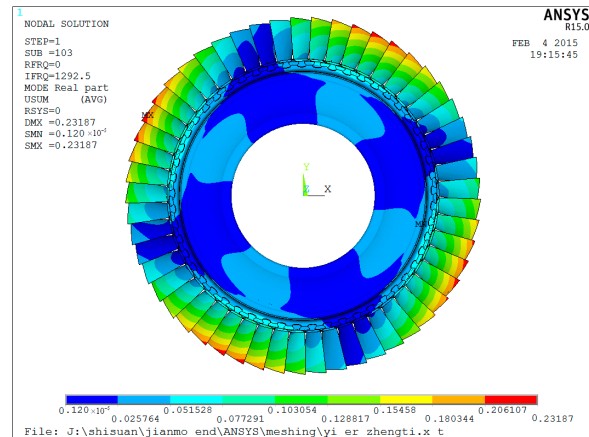

(**b**) Grade two leaf disk

**Figure 30.** Mode $99 \times 10^{-5} \times 10^{-4} \times 10^{-3}$.

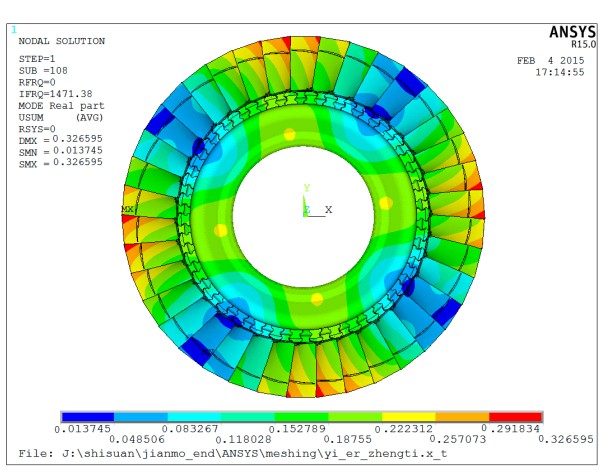

(**a**) Grade one leaf disk

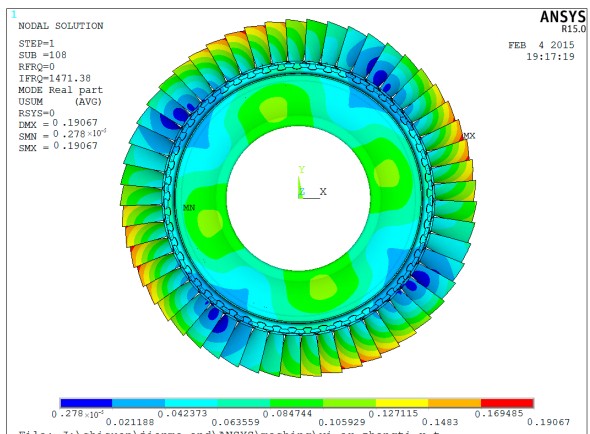

(**b**) Grade two leaf disk

**Figure 31.** Mode 104.

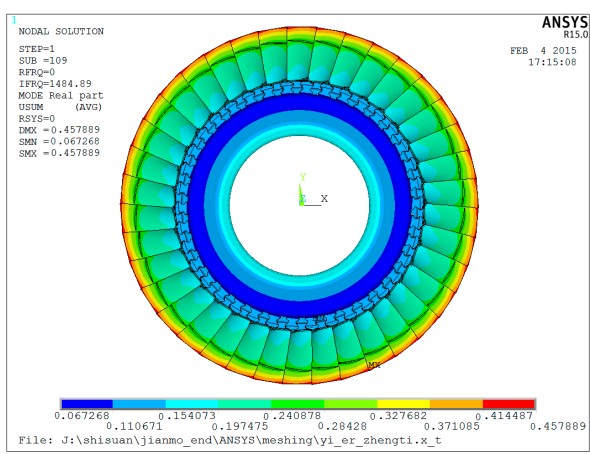

(**a**) Grade one leaf disk

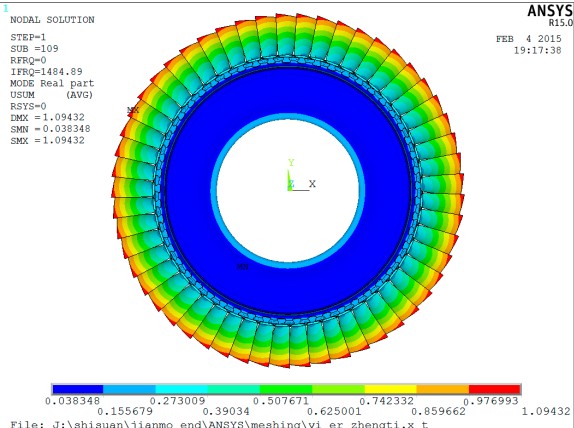

(**b**) Grade two leaf disk

**Figure 32.** Mode 105.

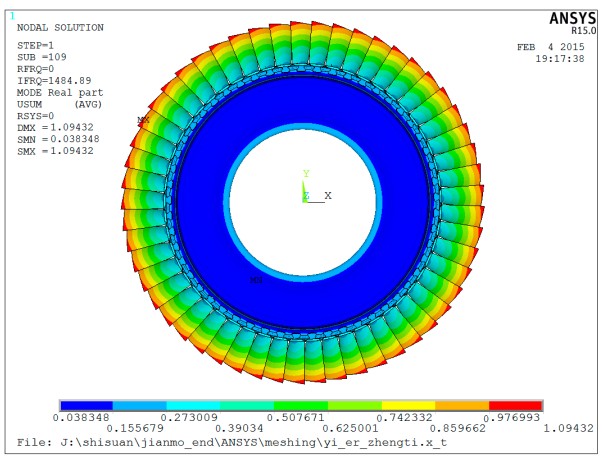 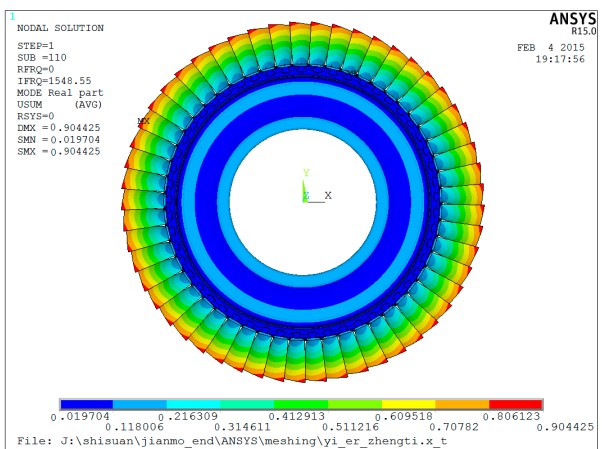

(**a**) Grade one leaf disk      (**b**) Grade two leaf disk

**Figure 33.** Mode 106.

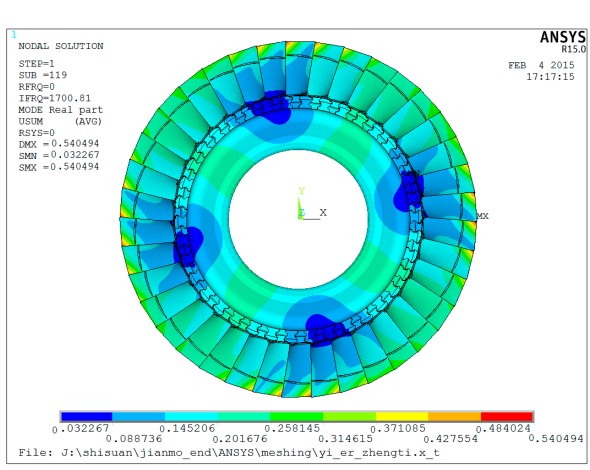 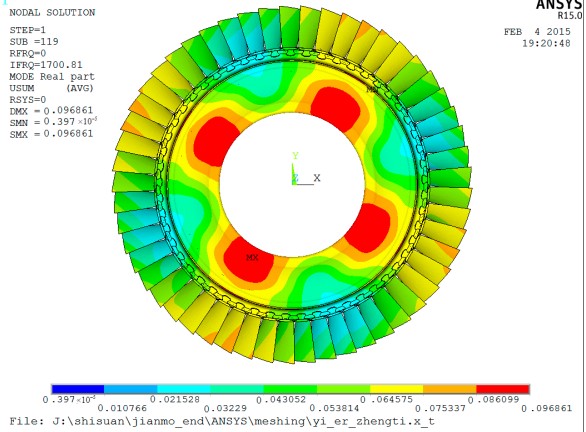

(**a**) Grade one leaf disk      (**b**) Grade two leaf disk

**Figure 34.** Mode 115.

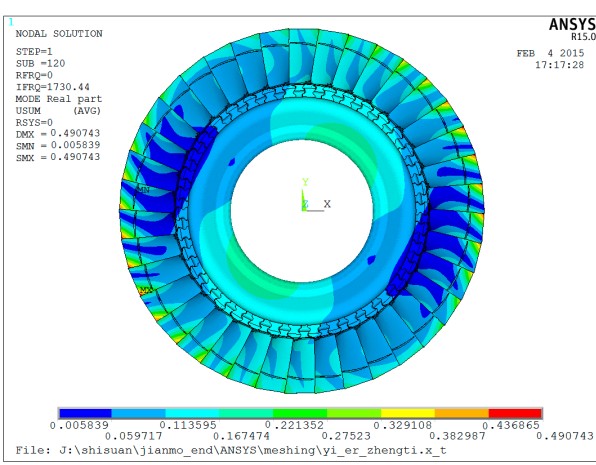

(**a**) Grade one leaf disk      (**b**) Grade two leaf disk

**Figure 35.** Mode 116.

*3.5. Comparison of the Results between the Two-Stage and Single-Stage Systems*

Table 11 shows a comparison of the natural frequencies and mode shapes of the two-stage bladed disk system with the first- and second-stage bladed disk systems.

**Table 11.** Comparison of natural frequencies between the two-stage and single-stage bladed disk coupled systems.

| Level One | | Level Two | | Two Levels of Coupling | | Vibration Mode |
|---|---|---|---|---|---|---|
| Order Number | Natural Frequency | Order Number | Natural Frequency | Order Number | Natural Frequency | |
| — | — | 1 | 650.63 | 1 | 656.78 | Coupled vibration of the two-stage bladed disk |
| 1–38 | 667.68–695.48 | — | — | 2–5 | 664.55–668.81 | Level one blade vibration |
| | | | | 6 | 674.13 | Coupled vibration of the two-stage bladed disk |
| | | | | 7–39 | 675.66–695.23 | Level one blade vibration |
| — | — | 2–53 | 706.22–749.50 | 40–91 | 720.99–749.30 | Level two blade vibration |
| 39–54 | 1064.10–1784.70 | 54–78 | 797.09–2048.40 | 92–103 | 778.35–1426.50 | Single stage coupled vibration of the bladed disk |
| | | | | 104–106 | 1471.40–1548.50 | Coupled vibration of the two-stage bladed disk |
| | | | | 107–114 | 1555.20–1695.50 | Single-stage coupled vibration of the bladed disk |
| | | | | 115–116 | 1700.80–1730.40 | Coupled vibration of the two-stage bladed disk |

According to the comparison of the natural frequencies and modes of the two-stage bladed disk system and the single-stage bladed disk system in Table 11, it can be seen that the vibration of the bladed disk system has a certain order: the blade vibration is first, followed by the coupled vibration of the bladed disk. At the intersection of the two stages, the interstage coupled vibration of the bladed disk will occur. It can be seen that the natural frequencies and mode shapes of the two-stage bladed disk coupled system not only include the natural frequencies and mode shapes of the first- and second-stage bladed disk, respectively, but also have the coupled modes of the two-stage bladed disk. Therefore, the two-stage bladed disk system model should be chosen to calculate the coupled vibration modes of the two-stage bladed disk system. For the dominant vibration mode of a single-stage disk, a single-stage bladed disk calculation model should be selected.

## 4. Conclusions

The whole coupled vibration mode of a two-stage bladed disk system was analyzed. Firstly, the modal analysis of the first- and second-stage bladed disks was carried out, and then the modes and configurations of the first- and second-stage bladed disks were solved, respectively. Finally, the modal analysis of the two-stage bladed disk coupled system was carried out, and the coupled vibration forms of the two-stage bladed disk were analyzed. The following conclusions were drawn.

(1) For the single-stage bladed disk system, the low-order mode shape is the first-order bending vibration of the blade according to the pitch diameter. With an increase in the mode order, the vibration of the disk is excited, resulting in the coupled vibration of the blade and the disk. As the modal order continues to increase, the blade begins to transform from a bending vibration to twisting vibration. At the same time, due to the coupling of the blade and the disk, the first-order bending frequency of the blade is increased.

(2) For the two-stage bladed disk coupled system, the vibration of the bladed disk has a certain order: the blade vibration is first, followed by the coupled vibration of the bladed disk. For the intersecting position of the two-stage bladed disk frequencies, the interstage coupled vibration of the bladed disk will occur, and the number of pitch diameters or pitch circles of the vibration would be the same. The natural frequencies and mode shapes of the two-stage bladed disk coupled system not only include the natural frequencies and mode shapes of the first- and second-stage bladed disks, respectively, but also have the coupled

modes of the two-stage bladed disk, so the multistage bladed disk model is more accurate to analyze.

**Author Contributions:** Conceptualization, Y.Y. and T.Z.; methodology, Y.Y. and T.Z.; software, Y.F.; validation, Y.Y. and T.Z.; formal analysis, X.J.; investigation, X.J.; data curation, Y.F.; writing–original draft preparation, Y.Y.; writing–review and editing, Y.Y., X.J., T.Z. and Y.F.; visualization, Y.F.; supervision, T.Z.; project administration, T.Z. All authors have read and agreed to the published version of the manuscript.

**Funding:** This research was funded by the National Natural Science Foundation of China (No. 52165014, No. 51805076, No. U1708255 and No. 51775093), National Science and Technology Major Project of China (J2019-I-0008-0008), and the Fundamental Research Funds for the Central Universities of China (N2105013).

**Conflicts of Interest:** The authors declared no conflict of interest.

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
