# Peer review of "Study on the Coupled Vibration Characteristics of a Two-Stage Bladed Disk Rotor System"

_applsci, doi:10.3390/app11188600_

Round 1

Reviewer 1 Report

The paper presents a comprehensive analysis of a large number of initial modes, by comparing single-stage independent analyses, and two-stage coupled analysis. The overall conclusion is obviously that the coupled analysis is more effective, ans so in principle always required. The comparison of all the modes and their evolutions is of interest anyway.
However, there are some specific issues that need to be considered, as listed below:
1. In the text there are several typos and errors, such as: 'behaciors' -> 'behaviours', 'reuslts' -> 'results', and several errors as space not introduced after the period, such as: 'appear.In addition'.
2. The Introduction is too long and mostly it is a list of the several contributions, and several self-citation can be found in the final part of the paragraph. This is in principle correct, but a shorter and a more meaningful introduction is suggested.
3. Are the proposed properties introduced in 'Materials and Methods' simulating two Titanium alloys?
4. [major] The reported equations are not so useful to introduced the superelement and the substructure methods. Graphical description could be much more effective.
5. In Table 1 the use of the dimensionless frequencies is misleading, also considering that the frequencies reported after are not as dimensionless anymore.
6. In Table 1 all the method errors are basically the same, so this error is systematic. Is there a possible explanation for this?
7. Usually the expression 'nodal diameters' is used instead of 'pitch diameters'.
8. Table 3. It is said that the table reports the first 10 order natural frequencies, while only 4 are reported.
9. [major] Table 4 and and similar other below should be put in a graphical form.
10. Figs. 13, 14 could be put together as Fig. 13 (a) and (b).
11. Table 8. The modes shown are 195 while in the text it is reported 200.
12. [major] Page 17. The way the modes are introduced: '1) Coupled mode shape when...', 2) ..., 3) ... is not clear. A more direct comparison should be provided.
13. Conclusions. The expression 'the modes and modes...' is wrong and repeated a few times.
14. [major] The second part of the paper does not seem so well connected with the first, because the two modelling techniques introduced above, the superelement and substructure methods, are not mentioned anymore. So, are the presented modal results obtained with this modal synthesis methods or just with a full model?

Author Response

Dear Editors and reviewers,

Re: Manuscript ID: 1374101 and Title: Study on coupled vibration characteristics of two-stage bladed disk rotor system

Thank you for your letter and the reviewers’ comments concerning our manuscript entitled “1374101” (ID). Those comments are valuable and very helpful. We have read through comments carefully and have made corrections. Based on the instructions provided in your letter, we uploaded the file of the revised manuscript. Revisions in the text are shown using red highlight for additions, and strikethrough font for deletions. The responses to the reviewer's comments are marked in red and presented following.

We would love to thank you for allowing us to resubmit a revised copy of the manuscript and we highly appreciate your time and consideration.

Sincerely.

Zhao.

Reviewer #1:

Q1. In the text there are several typos and errors, such as: 'behaciors' -> 'behaviours', 'reuslts' -> 'results', and several errors as space not introduced after the period, such as: 'appear.In addition'

Response: The grammatical and formatting errors in the text have been corrected.

Q2. The Introduction is too long and mostly it is a list of the several contributions, and several self-citation can be found in the final part of the paragraph. This is in principle correct, but a shorter and a more meaningful introduction is suggested.

Response: The introduction has been shortened, and the expression of self-cited articles in the introduction has been simplified.

Q3. Are the proposed properties introduced in 'Materials and Methods' simulating two Titanium alloys?

Response: The two material properties are for the material of blade tenon and tenon grooves. The inaccuracy in the original text has been modified.

Q4. [major] The reported equations are not so useful to introduced the superelement and the substructure methods. Graphical description could be much more effective.

Response:The analysis process of the substructure modal synthesis superelement method is supplemented in the article.

Q5. In Table 1 the use of the dimensionless frequencies is misleading, also considering that the frequencies reported after are not as dimensionless anymore.

Response:The dimensionless processing in Table 1 is to compare the errors between the two methods. In the following text, in order to be consistent with the frequencies in the various order modal graphs, no dimensionless processing is done.

Q6. In Table 1 all the method errors are basically the same, so this error is systematic. Is there a possible explanation for this?

Response: The explanation is supplemented in the article. (Since the number of modes intercepted by the substructures is the same, and the same finite element mesh model is used, the errors of the two methods at each frequency are relatively consistent.)

Q7. Usually the expression 'nodal diameters' is used instead of 'pitch diameters'.

Response:The expression ' pitch diameters ' have been replaced by ' nodal diameters ' in the text.

Q8. Table 3. It is said that the table reports the first 10 order natural frequencies, while only 4 are reported.

Response:The wrong statement has been revised in the text.

Q9. [major] Table 4 and and similar other below should be put in a graphical form.

Response:Table 4 has been put in a graphical form.

Q10. Figs. 13, 14 could be put together as Fig. 13 (a) and (b).

Response:Figs. 13, 14 have been put together as Fig. 13 (a) and (b).

Q11. Table 8. The modes shown are 195 while in the text it is reported 200.

Response:The wrong statement has been revised in the text.

Q12. [major] Page 17. The way the modes are introduced: '1) Coupled mode shape when...', 2) ..., 3) ... is not clear. A more direct comparison should be provided.

Response:The graphs with inconspicuous contrast were deleted, and the Two-stage bladed disk was marked in the existing graphics to improve the recognition of the two-stage coupled bladed disk shape.

Q13. Conclusions. The expression 'the modes and modes...' is wrong and repeated a few times.

Response:The wrong statement has been revised in the text.

Q14. [major] The second part of the paper does not seem so well connected with the first, because the two modelling techniques introduced above, the superelement and substructure methods, are not mentioned anymore. So, are the presented modal results obtained with this modal synthesis methods or just with a full model?

Response:In response to the above problems, an explanation has been supplemented at the beginning of the second part of the manuscript.

Reviewer #2:

Q1. The name of superelement approach is inconsistent in the paper, superelement analysis, superunit analysis, for example. The author should unify them.

Response:The name of superelement approach is uniformly expressed as superelement analysis in the article.

Q2. The modal synthesis superelement method is introduced in the first part(Introduction) and the second part(Materials and Methods). What is the difference between the two parts?

Response:The description of the two methods is unified into the second part, and supplements are made at the beginning of the second part for the connection between the first part and the second part.

Q3. The author should indicate that which stage of the blade-disk was verified In the 3.1 part.

Response:The 3.1 part is the analysis by selecting the first stage bladed disk system, which has been supplemented at the beginning of the second part of the text.

Q4. Why not consider the contact between rabbet and tongue-and-groove in the models?

Response:In the modal analysis of the two methods, the contact is regarded as a bound state, so the two methods cannot consider the contact nonlinearity.

Q5. the detuned bladed disk was mentioned many times in this paper, but there is no analysis on the detuned factors. It should be clearly stated whether the detuned blade is considered in the model.

Response:There is no analysis and discussion of detuning in the article, and the relevant expressions.

Reviewer 2 Report

This paper investigated in detail the coupled vibration characteristics of two-stage bladed disk rotor system. It is well-organized by style and content. I suggest minor revision before accepted. Comments are as follows.

1. The name of superelement approach is inconsistent in the paper, superelement analysis, superunit analysis, for example. The author should unify them.

2. The modal synthesis superelement method is introduced in the first part(Introduction) and the second part(Materials and Methods). What is the difference between the two parts?

3. The author should indicate that which stage of the blade-disk was verified In the 3.1 part.

4. Why not consider the contact between rabbet and tongue-and-groove in the models?

5. the detuned bladed disk was mentioned many times in this paper, but there is no analysis on the detuned factors. It should be clearly stated whether the detuned blade is considered in the model.

Author Response

(The authors gave the same response as above.)

Round 2

Reviewer 1 Report

The issues raised by the reviewers were all addressed, and corrections provided by the authors. The clarity of the paper is improved in this revised version.
However, I suggest to improve the quality of figures (a) from fig. 14 to fig. 28 which appear with very poor resolution.